



# Role of emission sources and atmospheric sink on the seasonal cycle of CH$_4$ and $\delta^{13}$-CH$_4$: analysis based on the atmospheric chemistry transport model TM5

Vilma Kangasaho[1], Aki Tsuruta[1], Leif Backman[1], Pyry Mäkinen[2], Sander Houweling[3,4], Arjo Segers[5], Maarten Krol[6,7], Ed Dlugokencky[8], Sylvia Michel[9], James White[9], and Tuula Aalto[1]

[1]Finnish Meteorological Institute, P.O. Box.503, FI-00101, Helsinki, Finland
[2]Aalto University, Finland
[3]SRON Netherlands Institute for Space Research, the Netherlands
[4]Vrije Universiteit Amsterdam, the Netherlands
[5]Netherlands Organisation for Applied Scientific Research (TNO), Utrecht, the Netherlandss
[6]Wageningen University & Research, Meteorology and Air Quality, the Netherlands
[7]IMAU, Utrecht University, the Netherlands
[8]NOAA Global Monitoring Laboratory (GML), USA
[9]NSTAAR, University of Colorado, USA

**Correspondence:** Vilma Kangasaho (vilma.kangasaho@fmi.fi)

**Abstract.** This study investigates the contribution of different CH$_4$ sources to the seasonal cycle of $\delta^{13}$C during years 2000–2012 using the TM5 atmospheric transport model. The seasonal cycles of anthropogenic emissions from two versions of the EDGAR inventories, v4.3.2 and v5.0 are examined. Those includes emissions from Enteric Fermentation and Manure Management (EFMM), rice cultivation and residential sources. Those from wetlands obtained from LPX-Bern v1.4 are also examined in addition to other sources such as fires and ocean sources. We use spatially varying isotopic source signatures for EFMM, coal, oil and gas, wetlands, fires and geological emission and for other sources a global uniform value.

We analysed the results as zonal means for 30° latitudinal bands. Seasonal cycles of $\delta^{13}$C are found to be an inverse of CH$_4$ cycles in general, with a peak-to-peak amplitude of 0.07–0.26 ‰. However, due to emissions, the phase ellipses do not form straight lines, and the anti-correlations between CH$_4$ and $\delta^{13}$C are weaker (-0.35 to -0.91) in north of 30° S. We found that wetland emissions are the dominant driver in the $\delta^{13}$C seasonal cycle in the Northern Hemisphere and Tropics, such that the timing of $\delta^{13}$C seasonal minimum is shifted by ~90 days in 60° N–90° N from the end of the year to the beginning of the year when seasonality of wetland emissions is removed. The results also showed that in the Southern Hemisphere Tropics, emissions from fires contribute to the enrichment of $\delta^{13}$C in July–October. In addition, we also compared the results against observations from the South Pole, Antarctica, Alert, Nunavut, Canada and Niwot Ridge, Colorado, USA. In light of this research, comparison to the observation showed that the seasonal cycle of EFMM emissions in EDGAR v5.0 inventory is more realistic than in v4.3.2. In addition, the comparison at Alert showed that modelled $\delta^{13}$C amplitude was approximately half of the observations, mainly because the model could not reproduce the strong depletion in autumn. This indicates that CH$_4$ emission magnitude and seasonal cycle of wetlands may need to be revised. Results from Niwot Ridge indicate that in addition to biogenic emissions, the proportion of biogenic to fossil based emissions may need to be revised.





## 1  Introduction

Methane ($CH_4$) is a greenhouse gas of which the abundance is severely perturbed by anthropogenic activities. It causes 28 times more radiative forcing than equal emissions of $CO_2$ when integrated over 100 years. The abundance of $CH_4$ in the atmosphere has more than doubled since the pre-industrial times (Hartmann et al., 2013). $CH_4$ is emitted to the atmosphere from thermogenic, pyrogenic, and biogenic sources, which can be of natural or anthropogenic in origin (Saunois et al., 2020).

Most of the seasonal cycle in $CH_4$ emission is driven by the pyrogenic and biogenic sources, such as biomass burning, wetlands and rice cultivation (Crippa et al., 2020; Bergamaschi et al., 2018; Bloom et al., 2017; Basso et al., 2016; Giglio et al., 2013). The processes are highly dependent on climatological and meteorological conditions, such as temperature and precipitation, and cultivation cycles. In contrast, thermogenic sources, such as fossil fuel extraction and distribution, have little month-to-month variation, although winter emissions may be greater in some regions due to consumption of natural gas for heating

(Crippa et al., 2020). Likewise, instantaneous perturbations in emissions may occur due to blowout events from natural gas wells (Kuze et al., 2020; Pandey et al., 2019).

Seasonal variations in wetland $CH_4$ emissions have been much studied by site-level measurements (e.g. Delwiche et al., 2021; Villarreal and Vargas, 2021), process-based land surface ecosystem models (e.g. Parker et al., 2020), and atmospheric inversions (e.g. Xu et al., 2016; Zhang et al., 2021), but there are still large uncertainties in the magnitude and timing of

maximum emissions on continental to regional scales (Warwick et al., 2016; Bergamaschi et al., 2018; Tsuruta et al., 2019). Anthropogenic based thermogenic and biogenic $CH_4$ emission cycles mainly depend on political decisions. Although some countries report emission magnitudes to e.g. UNFCCC, often only annual values are reported, and emissions from e.g. rice paddies may not properly consider e.g. temperature dependencies and soil properties (Yan et al., 2009). In addition, emissions from livestock (e.g. enteric fermentation and manure management) may have seasonal cycles depending on temperature (Els-

gaard et al., 2016). However, again, such information is often not included in the reported emissions, and only few global inventories take the seasonal changes from this sector into account (Crippa et al., 2020, and references therein).

$CH_4$ has two stable carbon isotopes, $^{12}C$ and $^{13}C$, and hydrogen isotopes, $^{1}H$ and $^{2}H$. For the carbon isotopes, their process specific isotopic signatures ($^{13}C/^{12}C$ ratio compared to a reference, denoted as $\delta^{13}C$) depend on processes that produce $CH_4$ (Nisbet et al., 2016). Generally, emissions with pyrogenic origin are most enriched in $^{13}C$, followed by the thermogenic sources.

Sources from biogenic origin are most depleted in $^{13}C$ (e.g. Nisbet et al., 2016; Sherwood et al., 2017). Such information has shown to be useful in quantifying $CH_4$ source distributions (Schwietzke et al., 2016; Thompson et al., 2018; Monteil et al., 2011; Lan et al., 2021), in addition to $CH_4$-only atmospheric inversions, which estimates total $CH_4$ budgets (e.g. Saunois et al., 2020; Houweling et al., 2014). However, the $CH_4$ flux information derived using the information from isotopic measurements still have large uncertainty as the isotopic measurements are still limited in both spatial and temporal coverage, and partly

overlapping signatures makes source division uncertain (Schwietzke et al., 2016). On top of that, the isotopic signature of emissions can vary significantly by locations due to differences in production processes, types of origin or methanogeneisis (Ganesan et al., 2018; Feinberg et al., 2018; Etiope et al., 2019; Brownlow et al., 2017). Ganesan et al. (2018) also warned that the emission quantification, including its seasonality, may lead to erroneous results without fully incorporating detailed spatial





information of the isotopic signatures. In addition, the fractionation factor of OH vary between studies (Saueressig et al., 2001;
Cantrell et al., 1990). The role and magnitude of tropospheric Cl sink is also uncertain ranging from 13–37 Tg CH$_4$ yr$^{-1}$ (Allan
et al., 2007) to 12—13 Tg CH$_4$ yr$^{-1}$ (Hossaini et al., 2016) to even smaller estimates (Gromov et al., 2018)].

The seasonal cycle of $\delta^{13}$C is determined by atmospheric sinks and emissions. Sinks enrich the atmosphere in the $^{13}$CH$_4$,
and they have a strong seasonality due to their kinetic isotopic effect (KIE). In general, the $\delta^{13}$C cycle is mirroring the CH$_4$
cycle in high southern latitudes, where the effect of emissions are small (Saueressig et al., 2001; Bergamaschi et al., 1996).
However, the $\delta^{13}$C cycle is known to be affected by the seasonal variations in emissions, where majority of the emissions
deplete the atmosphere in $^{13}$CH$_4$. Therefore, the $\delta^{13}$C cycles do not correlate well with the atmospheric CH$_4$ cycles, especially
in the Northern Hemisphere (NH) (Allan et al., 2001b; Bergamaschi et al., 2000; Tyler et al., 2007). These studies found
that the $\delta^{13}$C seasonal cycle reaches its maximum approximately two months later than CH$_4$ reaches its minimum in the NH.
Studies using inverse transport modelling indicate that the weak negative correlations and phase shifts are strongly influenced
by wetlands in the northern high latitudes and biomass burning in the Tropics (Bergamaschi et al., 2000; Allan et al., 2001b;
Fujita et al., 2018; Warwick et al., 2016).

In this study, we examine the average $\delta^{13}$C seasonal cycle for the time period 2000–2012, and CH$_4$ source and sink contri-
butions at 30° latitudinal bands based on the TM5 global atmospheric transport model. We estimate atmospheric CH$_4$ and $\delta^{13}$C
cycles using the most recent isotopic signatures published, and five sectoral emission fields with different seasonality, includ-
ing those from the anthropogenic sources, and examine the differences in the combined CH$_4$ and $\delta^{13}$C cycles. In addition, we
evaluate the seasonal cycle in anthropogenic emission by comparing model estimates derived using the two recent versions of
EDGAR inventory, v4.3.2 and v5.0, against observations from the National Oceanic and Atmospheric Administration Global
Monitoring Laboratory (NOAA/GML) and the Institute of Arctic and Alpine Research (INSTAAR).

## 2 Materials and Methods

### 2.1 TM5 atmospheric chemistry transport model

TM5 (Krol et al., 2005) is a global Eulerian atmospheric chemistry transport model. It is driven by ECMWF ERA-Interm
meteorological fields, which for this study is run on a 1° × 1° (latitude × longitude) zoom grid over Europe (up to 74° N)
embedded in a 4° × 6° global grid with an intermediate 2° × 3° zoom region (e.g. Tsuruta et al., 2017). Vertically, 25 layers are
used that are a coarsening from the original 60 ERA-Interim layers, and vertical mixing was calculated based on the Gregory
et al. (2000) convection scheme as archived in the ERA-Interim meteorological fields.

In this study, CH$_4$ (incl. $^{12}$CH$_4$ and $^{13}$CH$_4$) and $^{13}$CH$_4$ are transported as two separate tracers, and $\delta^{13}$C$-$CH$_4$ ($\delta^{13}$C) is
calculated as:

$$\delta^{13}\text{C}-\text{CH}_4 = \left( \frac{(^{13}\text{C}/^{12}\text{C})_{\text{sample}}}{R_{\text{std}}} - 1 \right) \times 1000, \tag{1}$$

where $R_{\text{std}} = 0.0112372$ is the isotopic ($^{13}$C and $^{12}$C) ratio of the standard, Vienna Pee-Dee Belemnite (VPDB; de Laeter et al.,
85 2003).





The atmospheric sink in TM5 includes off-line chemistry; photochemical reactions with OH, Cl, and O($^1$D). The reaction with OH, the largest sink of atmospheric CH$_4$, is calculated based on Houweling et al. (2014). The monthly variations in OH concentrations are based on Spivakovsky et al. (2000), scaled by 0.92 based on an evaluation using methyl chloroform (Huijnen et al., 2010). The first order loss rates for the reactions with Cl and O($^1$D) are considered only in stratosphere and are calculated separately, where the rates are based on ECHAM5 general circulation model (Jöckel et al., 2006). No interannual variation of the photochemical sink processes is included in this study, as interannual variations are often assumed to be small for the study period (Zhao et al., 2019; Turner et al., 2019; Rowlinson et al., 2019). Note also that the purpose of the study is to analyse the seasonal cycle, but not trends and interannual variations in the CH$_4$ and $\delta^{13}$C.

The kinetic isotopic effects (KIE) k($^{12}$CH$_4$)/k($^{13}$CH$_4$) = 1.004 is used for $^{13}$CH$_4$ reaction with OH (Crowley et al., 1999), and 1.066 and 1.013 are used for Cl and O($^1$D), respectively (Saueressig et al., 2001). In this study, the KIE of total CH$_4$ is assumed to be the same as for $^{12}$CH$_4$, i.e. k($^{12}$CH$_4$)/k($^{13}$CH$_4$) $\approx$ k(CH$_4$)/k($^{13}$CH$_4$).

In addition to the photochemical sinks, we include the sink to dry soils (i.e. a negative flux from atmosphere to soil) in the lowermost layer of TM5. CH$_4$ is oxidised by bacteria in aerobic mineral soils, and therefore, the sink is dependent on soil moisture, temperature and also soil texture (Spahni et al., 2011). These dependencies lead to the smallest sink in winter and largest sink in summer (Fig. 1). The soil sink can be treated as a pseudo first order reaction $L = k' \cdot [\text{CH}_4]$, where $k' = k/h$ and $h$ is the thickness of the lowermost layer. The flux $F$ at the soil surface is $F = k \cdot [\text{CH}_4]$. The $^{12}$CH$_4$ soil sink $F_{\text{soil},12}$ is taken from the LPX-Bern v1.4 land ecosystem process model (Lienert and Joos, 2018), and varies interannually and monthly. The removal rate of $^{12}$CH$_4$ is then $L_{\text{soil},12} = 1/h \cdot F_{\text{soil},12}$. The removal rate for $^{13}$CH$_4$ due to the soil sink, $L_{\text{soil},13}$, is therefore calculated as

$$L_{\text{soil},13} = \frac{F_{\text{soil},12}}{h \cdot \text{KIE}_{\text{soil}}} \times \frac{[^{13}\text{CH}_4]}{[^{12}\text{CH}_4]} \tag{2}$$

where $F_{\text{soil},12}$ is the negative flux of $^{12}$CH$_4$ at the surface, $h$ is the thickness of the lowermost layer, $[^{12}\text{CH}_4]$ and $[^{13}\text{CH}_4]$ are the atmospheric concentrations of $^{12}$CH$_4$ and $^{13}$CH$_4$, and KIE$_{\text{soil}}$ is assumed to be 1.0177 (Snover and Quay, 2000).

TM5 has been applied to various CH$_4$ studies, and initial 3-dimensional (3D) CH$_4$ fields were readily available from e.g. our previous study by Tsuruta et al. (2017). For $^{13}$CH$_4$, spin-up was needed to create 3D mixing ratio fields that are in approximate steady state. We run 40 years of spin-up (running TM5 using emissions and meteorological fields of year 2000 for 40 times), starting from the converted fields, and based on the emissions and isotopic signatures described in Sections 2.2 and 2.3. The spin-up was started by converting the well-mixed CH$_4$ fields to $^{13}$CH$_4$ fields based on Eq. 1, and assuming the average $\delta^{13}$C in the lowest model layer is -47 ‰ and that of the uppermost layer (95 Pa <) to be -30 ‰. During the spin-up, the spatial distribution and the shapes of vertical profile changed significantly. The value at uppermost layer increased much during the spin-up, and the stratospheric $\delta^{13}$C increased by ~20 ‰, reaching to approx. -10 ‰ at the end of the spin-up. The exact value of stratospheric $\delta^{13}$C is unknown due to lack of observations, but -10 ‰ is close to the previous studies (Röckmann et al., 2011; Saueressig et al., 2001). In this study, the focus is in the troposphere and the exact values in stratosphere are not important for our analysis.



## 2.2 CH$_4$ and $^{13}$CH$_4$ flux fields

The global CH$_4$ flux fields from anthropogenic and natural sources are taken from inventory and process-based model data (Sections 2.2.1 and 2.2.2). All the fields are pre-processed to a global $1° \times 1°$ grid to match the TM5 model resolution. $^{13}$CH$_4$ fluxes are calculated by converting CH$_4$ flux fields using isotopic signature (Table 1) and Eq. 1.

### 2.2.1 Anthropogenic CH$_4$ flux data

Monthly global anthropogenic emissions are taken from EDGAR inventories (https://edgar.jrc.ec.europa.eu) v4.3.2 (Janssens-
Maenhout et al., 2019) and v5.0 (Crippa et al., 2020)). The original resolution is $0.1° \times 0.1°$ (latitude $\times$ longitude) and the inventories are based on the geographical distribution of different activities e.g. energy, agricultural land use and traffic utilising GIS techniques.

The EDGAR database includes emissions from the IPCC 1996 classes 1, 2, 4, 6 and 7 listed in Supplementary Table S1. We categorised these classes into six components: enteric fermentation and manure management (EFMM), landfills and waste
water treatment (LWW), rice cultivation (RICE), coal, oil and gas, and residential (Supplementary Table S2). Among these, EFMM, LWW and RICE are anthropogenic biogenic sources, with depleted $\delta^{13}$C isotopic signature, while others are fossil-based sources which are enriched $^{13}$CH$_4$ (Fisher et al., 2017). The seasonal cycle of anthropogenic sources are dominated by "anthopogenic" biogenic sources - no significant seasonality is present in the fossil-based sources (Fig. 1).

V4.3.2 is the first EDGAR inventory to include seasonality. It provides monthly values for 2010. We calculated the seasonal
cycle for each $1° \times 1°$ grid by applying the 2010 seasonality to other years keeping the same annual totals. For v5.0, monthly values for year 2015 are available, and we applied its seasonality for each grid similarly to the procedure for v4.3.2.

The two EDGAR versions differ significantly in their seasonal cycles of EFMM and RICE (Fig. 1). For the global total EFMM emissions, v4.3.2 has a seasonal maximum in March and a minimum in November, and its average amplitude over 2000–2012 is 7.50 Tg CH$_4$ month$^{-1}$. In contrast, v5.0 has no seasonal cycle (Fig. 1, Supplementary Table S3). For the global
RICE emissions, v4.3.2 has its seasonal maximum in March and its minimum in January and December, and its average amplitude over years 2000–2012 is 2.61 Tg CH$_4$ month$^{-1}$. The v5.0 RICE emissions has its maximum in August and its minimum in March with an average amplitude over 2000–2012 of 5.51 Tg CH$_4$ month$^{-1}$ (Fig. 1, Supplementary Table S3). The differences in the seasonality is mostly due to the differences: in the spatial distributions in v4.3.2, the seasonality varies over some latitude bands, while in v5.0, it varies by country for which information is available (Crippa et al., 2020). In addition,
in v4.3.2 the same temporal profiles is used for all agricultural sectors, which is revised in v5.0 to better correspond each sector separately (Janssens-Maenhout et al., 2019; Crippa et al., 2020)

Annual totals of the two versions also differ slightly. The largest differences are in LWW, where v4.3.2 is lower than v5.0 by 8.2 Tg CH$_4$ year$^{-1}$ compared to v5.0. This gives a global total difference of $\sim 7$ Tg CH$_4$ year$^{-1}$ between the two versions (global totals for the year 2000 are 292.17 Tg CH$_4$ year$^{-1}$ and 299.05 Tg CH$_4$ year$^{-1}$ for v4.3.2 and v5.0, respectively). Both
EDGAR versions have similar trends dominated by increasing trend in EFMM, LWW, coal, and oil and gas (Supplementary

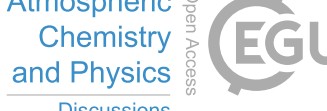

**Figure 1.** Monthly emission estimates for global (top) and latitudinal zonal totals. The left-hand side panel includes fossil-based emissions and the right-hand side panel emissions from biogenic origin, both natural and anthropogenic. Global emissions (top) are split into 30° latitude bands. Emissions from EDGAR v4.3.2 are for year 2010 and from EDGAR v5.0 for 2015. Natural sources are averages for 2000–2012. Shaded areas show minimum and maximum of the monthly totals over the 2000–2012.



Fig. S1). Note however, that this study focuses on the seasonal cycle, and that the analysis of trends will be presented in follow-up studies.

The 2000–2012 average annual total non-biogenic (coal, oil and gas, residential) emissions are similar in both versions, but the biogenic (EFWW, LWW, RICE) emissions are higher in v5.0. The mean ratios (biogenic/non-biogenic) are 1.89

(208.17/110.83 Tg $CH_4$ year$^{-1}$) and 1.97 (214.97/109.52 Tg $CH_4$ year$^{-1}$) in v4.3.2 and v5.0, respectively.

In addition to the two versions of the EDGAR emissions, we created emission fields based on v4.3.2, but removed the seasonal cycle of EFMM by taking annual means. This was used to test the effect of the seasonally varying EFMM emissions, which was largest in v4.3.2, but absent in v5.0.

### 2.2.2 Natural CH$_4$ flux data

Natural sources include those from wetlands, biomass burning, open ocean, termites and geological sources. Among these sectors, emissions from wetlands and termites are biogenic sources with depleted $\delta^{13}C$ values, while others are considered non-biogenic with more enriched $\delta^{13}C$ values (Table 1). Monthly wetland emissions are taken from the process-based land ecosystem model LPX-Bern v1.4 (Lienert and Joos, 2018), which is a dynamic vegetation model that estimates fluxes for wetlands. Wetland emissions have the largest seasonal cycle amplitude among all source categories (Fig. 1) with a global

average amplitude of 8.10 Tg $CH_4$ month$^{-1}$ averaged over 2000–2012. The seasonal minimum occurs in winter, and the maximum in summer, i.e. July - August in the NH and January–February in the Southern Hemisphere (SH) (Fig. 1).

Monthly biomass burning emissions are taken from GFED v4.2 (Giglio et al., 2013). Biomass burning emissions vary strongly from year to year, and the amplitude in the seasonal cycle varies much by year and locations (Fig. 1). For the global total, the average amplitude during 2000–2012 is 2.49 $CH_4$ Tg month$^{-1}$, and the maximum and minimum amplitudes during

2000–2012 are 3.95 and 1.07 $CH_4$ Tg month$^{-1}$, respectively. Monthly emissions from open ocean were calculated assuming a supersaturation of $CH_4$ in the seawater of 1.3 (Lambert and Schmidt, 1993). The sea-air flux of methane was calculated using ECMWF ERA-interim data (Dee et al., 2011) of sea surface temperature, sea ice concentration, surface pressure and wind speed (Tsuruta et al., 2017). The amplitude of its seasonal cycle is relatively small, with a global average of 0.08 $CH_4$ Tg month$^{-1}$ during 2000–2012. For termites and geological sources, no seasonality is taken into account. The emissions from

termites are taken from the VISIT process-based terrestrial ecosystem model (Ito and Inatomi, 2012), and gridded emission maps by Etiope et al. (2019) are used for geological sources. Geological emissions by Etiope et al. (2019) are scaled down from 37.4 Tg $CH_4$ year$^{-1}$ to 5 Tg $CH_4$ year$^{-1}$, based on Hmiel et al. (2020).

### 2.3 Isotopic signature

The global $^{13}CH_4$ flux fields were calculated from $CH_4$ emission fields and Eq. 1 using the isotopic signatures for each source

given in Table 1. For LWW, RICE, residential, ocean and termite emissions, the signatures are from Thompson et al. (2018) (mean values), and a single value is applied globally. Spatially varying isotopic signatures are used for EFMM, coal, oil and gas, wetlands, biomass burning and geological emissions. For EFMM, oil and gas, coal and biomass burning, we use the signatures from Feinberg et al. (2018) based on global chemistry-climate simulations of the SOCOL model. For coal, we use the M-COAL





**Table 1.** Isotopic signatures used to convert $CH_4$ flux fields to $^{13}CH_4$ fields. For values vary globally, ranges of values are shown. Please see Supplementary Fig. S2 for spatial distributions.

| Emission source | Signature value |
|---|---|
| Enteric Fermentation and Manure Management (EFMM) | [-67.9,-54.5][1], -66.8[2] |
| Landfills and Waste Water Treatment (LWW) | -55.6[2] |
| Rice (RICE) | -62.1[2] |
| Coal | [-64.1, -36.1][1], -40[2] |
| Oil and Gas | [-56.6, -29.1][1], -40[2] |
| Residential | -40[2] |
| Wetlands | [-74.9, -50][3], -61.3[2] |
| Fires | [-25, -12][1], -22.2[2] |
| Ocean | -47[2] |
| Termites | -65.2[2] |
| Geological | [-68, -24.3][4], -40[2] |

[1] Feinberg et al. (2018),[2] Thompson et al. (2018), [3] Ganesan et al. (2018), [4] Etiope et al. (2019)

version presented by Feinberg et al. (2018). For geological and wetland emissions, the signatures from Etiope et al. (2019) and
Ganesan et al. (2018) are used, respectively. The Ganesan et al. (2018) values are based on observations characterising wetland
ecosystems. The isotopic signatures from Feinberg et al. (2018) were originally given on in T42 resolution, and emission from
wetlands on 0.5° × 0.5°. We converted those to 1° × 1° resolution by choosing the closest coordinate value and by taking
simple grid averages, respectively. Grid cells with missing data are filled with mean values from Thompson et al. (2018). The
applied isotopic signatures do not have seasonal or inter-annual variations. This is appropriate if we assume that the spatial
distribution of the sources does not change, but only the magnitude.

We acknowledge that there are some differences in the spatial distributions of emissions used in e.g. Feinberg et al. (2018)
against the EDGAR versions and Ganesan et al. (2018) against LPX-Bern v1.4, i.e. the signatures are not custom-made for our
emission fields. Therefore, the corresponding signature values may not be appropriate in all grid cells. However, considering
the large range in source signatures (Schwietzke et al., 2016; Nisbet et al., 2016; Sherwood et al., 2017), we assume that our
values are a good approximation for examining the cause of $\delta^{13}C$ seasonal cycle.

### 2.3.1 Isotopic signatures in spin-up simulations

There are large uncertainties in the magnitude and spatial distribution of the isotopic signature, and therefore, we performed
several spin-up simulations with slightly different isotopic signature to examine the effect in $\delta^{13}C$ seasonal cycle. We first
examined the filled values (grids with no initial value assigned) by applying the values from Monteil et al. (2011) and Thompson
et al. (2018). We also used a weighted mean value, which lead to less negative values of $\delta^{13}C$, i.e. more enriched with $^{13}CH_4$
for most of the sources. In contrary to expectation, the different filling values did not affect the seasonality of $\delta^{13}C$, probably





**Table 2.** Set of simulations, anthropogenic emission fields used, and emission categories from which seasonal cycle is removed.

| Simulation | Anthropogenic emission fields | *Removed seasonal cycle |
| --- | --- | --- |
| SIM_E5 | EDGAR v5.0 | - |
| SIM_E5_WETNS | EDGAR v5.0 | Wetlands |
| SIM_NS | EDGAR v5.0 | All emissions |
| SIM_E432 | EDGAR v4.3.2 | - |
| SIM_E432_EFMMNS | EDGAR v4.3.2 | EFMM |

due to small emission magnitude in the regions, where the filling values were applied. In addition, we found that the simulated seasonal cycles in $^{13}CH_4$ are rather sensitive to the applied spatial distribution in source signature (e.g. decimal values instead of rounded integers), especially with region of high emission magnitude (Supplementary Fig. S3).

## 2.4 Atmospheric CH$_4$ and $\delta^{13}$C observations

We used CH$_4$ observations from National Oceanic and Atmospheric Administration - Global Monitoring Laboratory (NOAA/GML) and $\delta^{13}$C observations from Institute of Arctic and Alpine Research (INSTAAR), University of Colorado Boulder to evaluate the simulation results. In particular, we compared and evaluated model estimates against observations using data from Alert, Nunavut, Canada (ALT; 82.4508° N, 62.5072° W, CH$_4$: elevation = 185 m a.s.l., intake height = 5 m a.g.), Niwot Ridge, Colorado, USA (NWR; 40.0531° N, 105.5864° W, elevation = 3523 m a.s.l., intake height = 3 m a.g.) and South Pole, Antarctica (SPO; 89.98° S, 24.8° W, elevation = 2810 m a.s.l., intake height = 3–11.3 m a.g.). SPO is an optimal place to evaluate the seasonal cycle of background levels of CH$_4$, and there are no major CH$_4$ sources nearby. In contrast, NWR is located in the front range of the Colorado Rockies, and mainly measures well-mixed background air. NWR measurements influenced by strong anthropogenic sources are filtered out. Finally, ALT is located far away from anthropogenic sources, and samples air that is more influenced by distant wetland fluxes, whereas SPO is located far away from all sources both natural and biogenic. Note, none of the stations are located in the TM5 1° × 1° zoom region, and the model values are sampled from 4° × 6° grid using 3D linear interpolation.

For comparison, observations from 2002–2012 are used. The first two years were omitted from the analysis to be comparable to the modelled seasonality (see Section 3.1.1). To obtain detrended data we used curve fitting methods by Thoning et al. (1989). We calculated the short term smoothed curve and the trend curve. The detrended seasonal cycle is obtained by subtracting the trend curve from the smooth curve. The $\delta^{13}$C observations from 2007 onward have different trends than those in 2002–2006 (Nisbet et al., 2019). However, using the method from Thoning et al. (1989) we can compare years with different trends.

### 2.5 Simulation setups

We carried out five TM5 simulations using different input emission fields for 2000–2012 (Table 2). The end year 2012 is the last year for which the EDGAR 4.3.2 data is available. To examine the effect of the seasonal cycle in emissions, we used two





versions of the EDGAR inventory, v5.0 (SIM_E5) and v4.3.2 (SIM_E432), and those without EFMM seasonal cycle in v4.3.2 (SIM_E432_EFMMNS). In addition, we examined the seasonal cycle in wetland emission by using annual mean emissions (SIM_E5_WETNS) instead of a seasonal cycle. We further examined the $\delta^{13}$C seasonal cycle exclusively caused by the $CH_4$ sinks by removing the seasonal cycle of all emission sources (SIM_NS).

## 3   Results

### 3.1   Zonal means near the surface

Detrended zonal mean atmospheric $CH_4$ ($\Delta CH_4$) and $\delta^{13}$C ($\Delta\delta^{13}$C) values from the simulations are compared at 30° latitudinal bands. The trend and smoothed fit are calculated for 2000–2012 from the lowest five layers of TM5 (up to approx. 850 hPa) based on Thoning et al. (1989), and the detrended smoothed fit is averaged over 2002-2012 to examine the seasonal cycle. Note that we found that it takes approximately two years for the seasonal cycle of the lower atmosphere to stabilise by changing emission fields to those used in spin-up (Supplementary Fig. S5). Therefore, in order to remove the effect of initial state, the first two years of the forward simulations are omitted from the analysis. In this section, we focus on the seasonal cycle in $\Delta\delta^{13}$C and its relation to $\Delta CH_4$ cycle in Section 3.1.2, as $CH_4$ cycle has been discussed extensively in previous studies (e.g. Dlugokencky et al., 1997; Javadinejad et al., 2019; Kivimäki et al., 2019; Khalil and Rasmussen, 1983).

We acknowledge that the $\delta^{13}$C cycles are affected by local sources, and can vary spatially at smaller resolution than 30° latitudinal bands (Hein et al., 1997; Allan et al., 2001b; Warwick et al., 2016; Fujita et al., 2018; Bergamaschi et al., 2000). In addition, tropical meteorological dynamics such as the positions of the Intertropical Convergence Zone and the South Pacific Convergence Zone affect the seasonality of $CH_4$ and $\delta^{13}$C, and these variations can not be distinguished by using 30° latitudinal means (Lowe et al., 2004).

### 3.1.1   Peak-to-peak amplitude and shape of $\delta^{13}$C seasonal cycle

Generally, the seasonal cycle of $\Delta\delta^{13}$C mirrors the seasonal cycle of $\Delta CH_4$, such that $\delta^{13}$C has seasonal minimum in winter and maximum in summer in the NH, and vice versa for the SH (Fig. 2). Seasonal variations of both $CH_4$ and $\delta^{13}$C are larger in the NH than in the SH, mostly because the major emission sources are located in the NH. The seasonal cycle amplitude for $\Delta CH_4$ in our model is the largest in the NH Tropics EQ–30° N (49.7 ppb, SIM_E5), while that for $\Delta\delta^{13}$C is the largest at 60° N–90° N (0.26 ‰, SIM_E5). The smallest amplitude is found in the SH Tropics (30° S–EQ) for both $\Delta CH_4$ and $\Delta\delta^{13}$C, with 15.1 ppb and 0.07 ‰, respectively (SIM_E5). This is well in line with previous studies (e.g. Allan et al., 2001a; Tyler et al., 1994a). The $\Delta\delta^{13}$C seasonal cycle amplitude in 90°S–60°S approximately half of 60° N–90° N, while it is 25 % smaller in $\Delta CH_4$. Wetland emissions are the largest natural source of $CH_4$ and have the largest seasonal cycle among all emission categories, with the highest emissions during summer and autumn (in respective hemispheric seasons) (Fig. 1). Wetlands are biogenic sources, with depleted isotopic signatures (Section 2.3), i.e. increases in wetland emissions will result in decreases in $\delta^{13}$C. When the seasonality of wetland emissions is removed (SIM_E5_WETNS), the largest seasonal cycle amplitudes are found in 60° N–90°





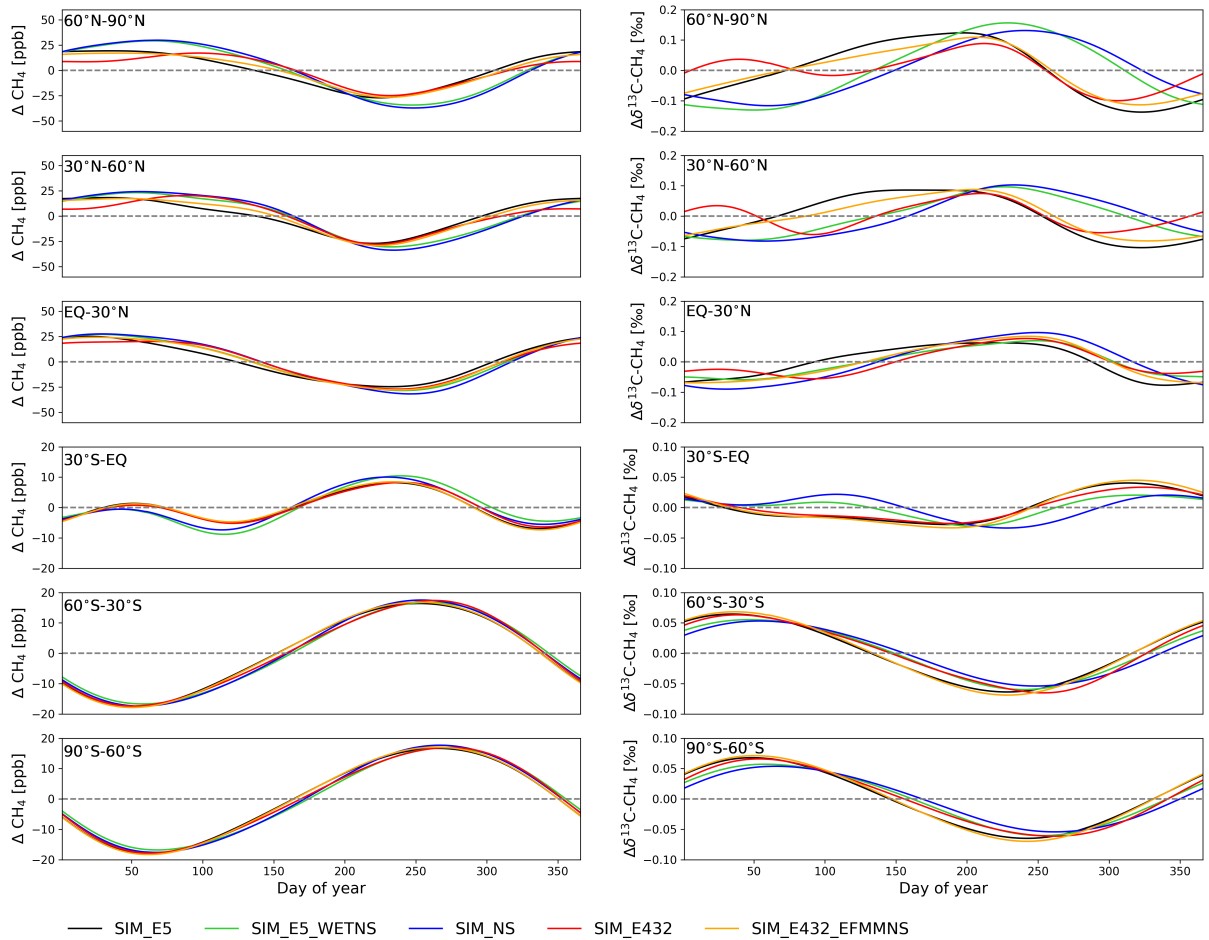

**Figure 2.** Detrended zonal mean averages for $CH_4$ (left) and $\delta^{13}C$ (right) from model simulations, averaged over 2002–2012 and the lowest 5 layers. Note differences in y-axis for the Northern and Southern Hemispheres.

N for both $\Delta CH_4$ (63.4 ppb) and $\Delta\delta^{13}C$ (0.29 ‰), which is an increase of 28 % and 10 % compared to SIM_E5, respectively. The smallest amplitudes are again found in 30° S–EQ (19.3 ppb and 0.05 ‰). In that latitude band, the $\Delta\delta^{13}C$ amplitude decreased 25 % compared to SIM_E5, and that of $\Delta CH_4$ shows an 28 % increase. Generally in the NH, the amplitude of $\Delta CH_4$

increased (12–37 %), while the $\Delta\delta^{13}C$ amplitudes decreased slightly at all latitudes (7–25 %), except for 60°N–90°N, which increased by 10 %. Although wetland emissions have the largest seasonal cycle amplitude in the Tropics (Fig. 1), removing the seasonal cycle resulted in an increase in the $\Delta CH_4$ amplitudes because a compensating effect is eliminated: normally wetland emissions increases (decreases) at the same time as the oxidation capacity increases (decreases). In contrast, the $\Delta\delta^{13}C$ amplitude at 60° N–90° N increased because normally wetland emissions decreased (increased) when OH concentrations are

high (low). The results also indicate that the contribution of the wetland emissions to the $\Delta\delta^{13}C$ amplitude is equally strong in mid-latitudes in both hemispheres (10 % in 60° S–30° S and 7 % in 30° N–60° N), while the effect on $\Delta CH_4$ is much stronger


in the NH (19 %, 30° N–60° N) compared to the SH ($\leq -2$ %, 30° S–60° S). The largest differences in the shapes of the $\Delta\delta^{13}$C seasonal cycles between SIM_E5_WETNS and SIM_E5 are found at latitudes north of 30° S (Fig. 2). In the NH, $\Delta\delta^{13}$C in SIM_E5 is increasing in spring towards autumn, while it is decreasing in SIM_E5_WETNS in spring. The depletion of $\Delta\delta^{13}$C

in autumn is more gradual in SIM_E5_WETNS, and continuously decreasing towards winter. The time when the seasonal minimum occurs is significantly shifted from the end of the year to the beginning of the year by 69–93 days in the NH. The changes in phase of peak maxima are smaller (6–32 days), except for the band 30° N–60° N, which is shifted towards autumn by 67 days. In 30° S–EQ, $\Delta\delta^{13}$C is decreasing in the beginning of year in SIM_E5, while it is increasing in SIM_E5_WETNS. This creates two maxima peaks in SIM_E5_WETNS; one on DOY = 98 and another on DOY = 319, although the latter peak

is more than two times larger (Fig. 2).In south of 30° S, the differences are small, but the timing of the minima and maxima peaks are consistently shifted later by ~9–12 days. Although wetland emissions in the NH increase in spring, the effect on $\Delta\delta^{13}$C is seen in autumn (lag-effect). When the effect of the emissions is removed, the $\Delta\delta^{13}$C cycle closer follows the cycle in atmospheric oxidation. When the seasonal cycles of all emissions (SIM_NS) are removed, i.e. the seasonality is only driven by the atmospheric and soil sinks (see Sections 2.1 and 2.2), the seasonal cycles of $\Delta$CH$_4$ and $\Delta\delta^{13}$C are affected similarly to

SIM_E5_WETNS (Fig. 2). This indicates that wetlands are the dominant source driving the $\Delta\delta^{13}$C seasonal cycle, apart from sinks. Similarly to SIM_E5_WETNS, $\Delta\delta^{13}$C in the NH decreased in spring and gradually decreased in autumn. The $\Delta\delta^{13}$C seasonal cycle amplitude in SIM_NS is smaller compared to SIM_E5 at all latitudes, except for EQ–30° N where the SIM_NS amplitude (0.19 ‰) is the largest among all simulations. The timing of minima shift to later date compared to SIM_E5_WETNS in general (Fig. 2). The most notable differences are found in 30°S–EQ, where the maximum in SIM_NS occurs on DOY =

108, while all other simulations have maxima between DOY 313 and 324. Together with the results from SIM_E5_WETNS, this indicates that wetland emissions contribute to the depletion of $\Delta\delta^{13}$C in the beginning of the year in this latitude band, and other emissions contribute to enrichment of $\Delta\delta^{13}$C at later days. This also indicates that there is only a small lag-effect unlike in the NH; wetland emissions with depleting isotopic signatures are high in January–March, and biomass burning with enriched signature values are high in August–October (Fig. 1). Changing the version of EDGAR inventory (SIM_E432 vs

SIM_E5) results in differences in the simulated seasonal cycles, mostly in the NH, as expected. Most notably, NH $\Delta\delta^{13}$C from SIM_E432 shows a clear depletion in spring; it is higher in the beginning of the year, increases for a month or two, and decreases significantly in spring (Fig. 2). The maxima are lower and the depletion in autumn is less significant (Fig. 2), resulting in smaller seasonal cycle amplitudes by 5–29 % in all latitude bands north of 30° S compared to SIM_E5. The seasonal cycle of anthropogenic biogenic sources is driven by EFMM and Rice. Although the emission amplitude of these sources is greater

in EDGAR v4.3.2 in 30° N–60° N, the $\Delta\delta^{13}$C amplitude is smaller. This is because EFMM and rice emissions in v4.3.2 are high in spring only one month, while for v5.0 rice emissions are high in summer and remain high for three months (Fig. 1). The minimum in SIM_E432 occur earlier by ~20 days in latitudes 60° N–90° N compared to SIM_E5. For 30° N–60° N, unlike SIM_E5, SIM_E432 has two minima peaks (DOY = 92 and DOY = 294), with spring peak being lower. The largest differences in the timing of the maxima are found in 30° N–60° N, where the SIM_E432 peak occurs 51 days later than that of SIM_E5.

It may look as if the effect of anthropogenic biogenic sources comes without much lag-effect, in contrast to wetland emissions (those emissions are larger in spring and lower in winter in EDGAR v4.3.2 than in v5; Fig. 1), but the $\Delta\delta^{13}$C maximum is ~28





% lower in SIM_E432 (north of 30° N). Less EFMM emissions in winter could explain the earlier minimum and higher $\Delta\delta^{13}$C at the end and beginning of the year compared to SIM_E5. Therefore, we suspect that the effect continues for a few months. It is also noted that, the effects of changes in emissions are clearly seen in 60° N–90° N, although anthropogenic emissions and

their seasonal cycle in that latitude band are small (Fig. 1). This indicates the strong effect of mid-latitude emissions to the high northern latitudes. When the EFMM seasonal cycle is removed (SIM_E432_EFMMNS), the $\Delta\delta^{13}$C seasonal cycle is closer to that of SIM_E5 than to SIM_E432. The spring depletion is not seen, and magnitude of autumn depletion is similar to SIM_E5 (Fig. 2). However, the maximum is lower and the minimum is higher than in SIM_E5, resulting in 11–15 % smaller amplitudes in latitudes north of 30° N. The amplitudes increase at all other latitudes compared to SIM_E5, but with smaller magnitudes

(6–15 %). No significant differences in the timing of minima and maxima are found at other latitudes compared to SIM_E5, except for the timing of the maximum at 30°N–60°N, which is 46 days later. Compared to SIM_E432, the maxima are higher, confirming that the high spring EFMM emissions result in lower $\Delta\delta^{13}$C in halfway of the year (June–August). In addition, the lower winter EFMM emissions contribute to an increase in $\Delta\delta^{13}$C in the beginning of the year, i.e. there is a small lag-effect in how emissions affect $\Delta\delta^{13}$C the cycle.

### 3.1.2  Phase ellipses

The seasonal cycle of $\Delta\delta^{13}$C with respect to the $\Delta$CH$_4$ cycle can be examined with a so-called phase ellipse (Bergamaschi et al., 2000; Allan et al., 2001b). In this study, we examine phase ellipses, where the detrended daily averages of CH$_4$ are plotted against that of $\delta^{13}$C. Fig. 3 shows phase ellipses from the simulations using different emission fields at 30° latitudinal bands. In SIM_NS, the seasonal cycle of the emission components are removed, and the $\Delta$CH$_4$ to $\Delta\delta^{13}$C ratio is only driven

by the sinks (atmospheric and soil sinks). The results show high eccentricity of the phase ellipse at south of 60° N, i.e. close to a line, and the correlation $p_0 \approx -1$ and $R^2 \approx 1$ (Fig. 3). The effect of the soil sink is small at these latitudes, so probably the seasonal cycle of $\Delta\delta^{13}$C is preliminarily driven by the atmospheric sinks at these latitudes. Note that the rotation with respect to the DOY on the NH is anticlockwise, and that on the SH clockwise (Fig. 3).

In addition, we examine the timing $d$ (DOY) when the shifted correlations ($p_s$) between $\Delta$CH$_4$ at time $t$ and $\Delta\delta^{13}$C at time

$t+d$ are at minimum and maximum. When there are seasonal cycles only in the atmospheric sinks, we expect $d_{\min} = 0$ and $d_{\max}$ = 366/2 = 183, with $p_s(d_{\min})$ = -1 and $p_s(d_{\max})$ = 1. We use $d_{\min} = d'$ and $d_{\max} = d''$ to denote the specific case when there are no surface emissions or sinks affecting the seasonal cycle. The shifts in $d_{\min}$ and $d_{\max}$ indicate the differences in $\Delta$CH$_4$ and $\Delta\delta^{13}$C cycles, such that the times when $\Delta\delta^{13}$C are in decreasing or increasing phases are earlier or later than phase of $\Delta$CH$_4$.

In SIM_NS, there is no notable shift in $d$ (Supplementary Fig. S4) except for $d_{\max}$ in 30° S–EQ and 60° N–90° N. In 30°

S–EQ, the phase ellipse is close to a straight line, with $p_0 \approx -1$ and $R^2 \approx 1$, but $d_{\max} = 116$, which is 67 days earlier than $d''$. There are actually two maxima in the shifted correlations, at DOY = 116 and 270, and $p_s$ on these days is ~0.5 (Supplementary Fig. S4). This corresponds mainly to the OH and temperature cycles (OH reaction rate is strongly depended on temperature), where temperature is at maximum in April, and OH concentration in September. In 60°N–90°N, the correlation $p_0$ = -0.98 and $R^2 = 0.96$ are not at the minimum or maximum (Fig. 3), and $d_{\max}$ is shifted approx. by -10 days (Supplementary Fig. S4). This

is probably due to the effect of the soil sink, which has a larger seasonal cycle amplitude in the NH (1.45 Tg CH$_4$ yr$^{-1}$) than in

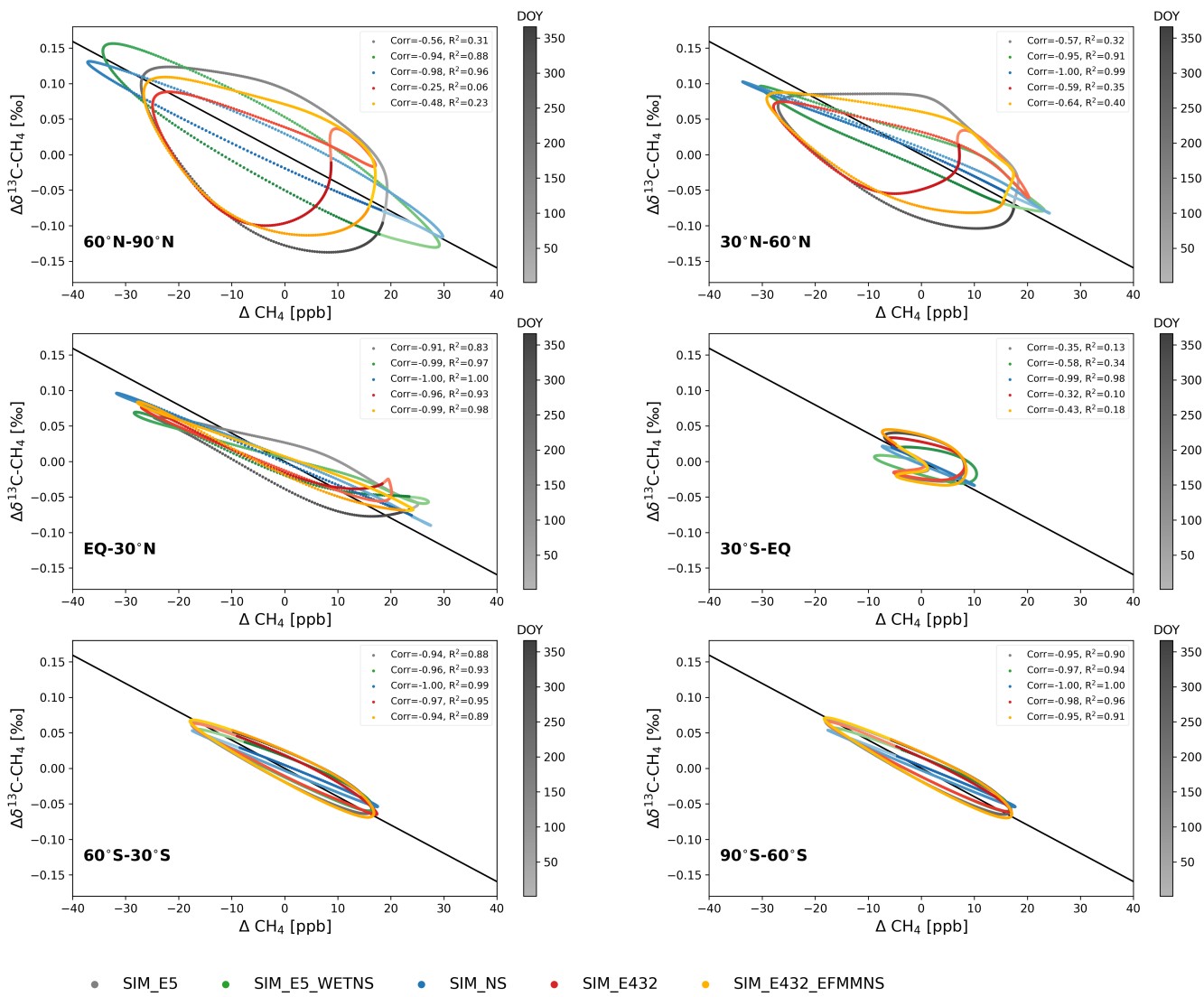

**Figure 3.** Detrended daily average CH$_4$ against $\delta^{13}$C from the lowest 5 levels over 2002–2012 at 30° latitudinal bands. Colour schemes indicate different simulations, and the colour darkness illustrates day of year (DOY). The lightest colours are DOY = 1 and the darkest DOY = 366. Solid black line is the KIE-line of SIM_NS, and considering only the OH sinks.





the SH (0.53 Tg $CH_4$ yr$^{-1}$) (2000), although the amplitude is not the largest in 60° N–90° N. The soil sink is included only at the lowest layer of TM5, and therefore, transport from lower latitudes and vertical mixing could affect the $\delta^{13}$C seasonal cycle in the upper levels and high latitudes. In addition, the so-called rectifier effect, i.e. correlations between emission and vertical transport, could result in slower $^{12}CH_4$ transport to the SH during the NH summer when soil sink is strongest. This then gives

higher $\delta^{13}$C value in the SH summer. In addition to the phase ellipses, Fig. 3 illustrates the theoretical KIE line when only the OH sink is considered (Allan et al., 2001b). The KIE-line slope is calculated as $\epsilon(1+\delta_0)$, where $\epsilon$ is $(k_{13}/k_{12})_{OH}-1$ and $\delta_0$ is the mean of $\Delta\delta^{13}$C in SIM_NS. This resulted in KIE of -3.98 ‰.Even in the high SH (90° S–60° S), where the effect of soil sink is small, the phase ellipse from SIM_NS does not exactly follow the KIE-line, indicating the effect of sinks other than OH, i.e. stratospheric Cl, O($^1$D) and soil sinks, and horizontal long-range and vertical transport.

When the emissions' seasonality is included, the ellipses' eccentricity decreases and the shape becomes more like a circle (Fig. 3). The correlation $p_0$ becomes weaker, R$^2$ smaller, and $d$ shifts by -97 days the most, indicating the differences in the shape of $\Delta CH_4$ and $\Delta\delta^{13}$C seasonal cycles. In SIM_E5, the correlation $p_0$ is weak (-0.57 to -0.56) and R$^2$ small (0.31–0.32) in north of 30° N (Fig. 3). The phase ellipses in those latitudes show that 1) both $\Delta\delta^{13}$C and $\Delta CH_4$ increase at the end and beginning of the year, i.e. there is no inverse relation in $\Delta\delta^{13}$C and $\Delta CH_4$ as in SIM_NS. 2) The relative rate of increase in

$\Delta\delta^{13}$C is slower compared to the relative rate of decrease in $\Delta CH_4$ in spring, and 3) the relative rate of decrease in $\Delta\delta^{13}$C is faster compared to the relative rate of increase in $\Delta CH_4$ in autumn. This effect creates a phase ellipse closer to an oval shape with one axis of symmetry (i.e. an egg-like shape), where the near-circle ellipses (i.e. smaller eccentricity) are formed at end and beginning of the year (short-half), and the other half of the ellipses with higher eccentricity (long-half) (Fig. 3). The $\Delta\delta^{13}$C seasonal cycle amplitude in 60° N–90° N is ~37 % larger than the amplitude at 30° N–60° N (0.26 and 0.19 ‰), creating a

larger circle at 60° N–90° N compared to 30° N–60° N (Fig. 3). At latitudes north of 30° N, $d_{\min}$ and $d_{\max}$ are shifted by approx. -55 and -63 days compared to SIM_NS (Supplementary Fig. S4). As Fig. 2 shows, this indicates that the minimum of $\Delta\delta^{13}$C is ~60 days earlier than the maximum of $\Delta CH_4$. In addition, the time of the increasing $\Delta\delta^{13}$C and decreasing CH$_4$ period differ by 39 and 16 days at 30°N–60°N and 60°N–90°N, respectively. This is the effect of changes in the ratio of biogenic and fossil based emissions. At EQ–30°N, the SIM_E5 phase ellipse is closer to a line, especially for the mid-year, compared to that of

north of 30° N (Fig. 3). Although eccentricity is smaller for the beginning and the end of the year (short-half), and $\Delta\delta^{13}$C and $\Delta CH_4$ are almost always negatively correlated at other days, with $d_{\min}$ and $d_{\max}$ shifts of only -24 days.

The SIM_E5 phase ellipse at 30° S–EQ forms an irregular shape, with weak negative correlation $p_0$ = -0.35 and R$^2$ = 0.13 (Fig. 3). As illustrated in Fig. 2, $\Delta\delta^{13}$C decreases in the beginning of the year until approx. DOY = 200, while $\Delta CH_4$ has both increasing and decreasing phases during that time. This creates a zigzag line for DOY ≤ 200; 1) close to the SIM_NS line at

approx. DOY ≤ 50, 2) perpendicular to the SIM_NS line at approx. 50 < DOY ≤ 120, and 3) horizontal line without much slope at approx. 120 < DOY ≤ 200 (Fig. 3). After approx. DOY ≥ 200, $\Delta CH_4$ and $\Delta\delta^{13}$C are anti-correlated, where the slope of the phase ellipse is close to that of SIM_NS. Compared to SIM_NS, the $p_s(d_{\max})$ is higher, with shift in $d_{\max}$ by -30 days, and the second $p_s$ peak is much smaller (Supplementary Fig. S4). In contrast, the correlation $p_s(d_{\min})$ is much weaker than the correlation of SIM_NS. In this latitude band, wetlands and biomass burning are the main contributors to the emission seasonal

cycle. Wetland emissions maximize in February–March, while biomass burning maximizes in September–October (Fig. 1).


Thus, although the $CH_4$ cycle is preliminary driven by OH, differences in the isotopic signatures of the emissions significantly affect the anti-correlation with $\delta^{13}C$ cycle.

At latitudes south of 30° S, the SIM_E5 phase ellipses' eccentricities are very high (Fig. 3) compared to the ellipses found in the NH. The strong correlation $p_0 \leq -0.94$ and $R^2 \geq 0.88$, with shifts in $d_{min}$ and $d_{max}$ by only ~-20 days (Supplementary Fig. S4), indicating that the $\delta^{13}C$ cycle with respect to the $CH_4$ cycle is preliminary driven by the sinks and little affected by the seasonal cycle of the emissions.

In SIM_E5_WETNS, the phase ellipses are closer to that of SIM_NS than to SIM_E5 at all latitudes except 30° S–EQ (Fig. 3). $\Delta\delta^{13}C$ and $\Delta CH_4$ are mostly anti-correlated in SIM_E5_WETNS, such that the eccentricity of the short- and long-halves of the ellipses do not differ much, and the correlations are strong ($p_0 \leq -0.94$) and $R^2$ are high ($\geq 0.88$ ) at all latitude except 30° S–EQ. This indicates that much of the $\Delta\delta^{13}C$ cycle with respect to $\Delta CH_4$ is driven by wetland emissions. Wetland emission has the largest seasonal cycle amplitude among the emission sources (Fig. 1), and is a main driver for the shape of the $\Delta\delta^{13}C$ cycle (see Section 3.1.1). However, for $\Delta CH_4$, the seasonal cycle is mainly driven by OH, such that the $\Delta CH_4$ cycles do not vary significantly by changes in the emission fields (Fig. 2). At 30° S–EQ the phase ellipse is closer to that of SIM_E5 than to SIM_NS (Fig. 3). The shape of the SIM_E5_WETNS $\Delta\delta^{13}C$ seasonal cycle in the beginning of the year is closest to that of SIM_NS, while it is closer to SIM_E5 in the second half of the year (Fig. 2). However, the $\Delta\delta^{13}C$ increasing and decreasing rates in the beginning of the year (~50 < DOY < 100 and 100 < DOY < 200) are smaller than that of SIM_NS (Fig. 2), creating the phase ellipse offset from the SIM_NS line (Fig. 3). This indicates that, at this latitude band, biomass burning emissions also contribute significantly to the $\Delta\delta^{13}C$ cycle with respect to $\Delta CH_4$ cycle.

The phase ellipses using EDGAR v4.3.2 (SIM_E432, SIM_E432_EFMMNS) generally form circles with smaller radius compared to those of SIM_E5 (Fig. 3) due to smaller $\Delta\delta^{13}C$ peak-to-peak amplitudes (Fig. 2). The shapes of SIM_E432_EFMMNS ellipses at north of 30° N are closer to those of SIM_E5 compared to SIM_E432. This is expected, as the $\Delta\delta^{13}C$ seasonal cycle is close to that of SIM_E5 (Section 3.1.1). The $\Delta\delta^{13}C$ cycle at EQ–30° N is closest to that of SIM_WETNS (Fig. 2), which is also reflected by the phase ellipse being closest to SIM_WETNS. The phase ellipses of SIM_E432 at latitudes north of 30° N are unique, such that they form an extra circle in the beginning of the year (~DOY < 75), outside the oval shape. This results in the weakest anti-correlation ($p_0$ = -0.25) and smallest $R^2$ = 0.06 at 60° N–90° N. These $p_0$ and $R^2$ statistics from SIM_E432 are among the smallest at other latitude bands as well.

### 3.2 Comparison to surface observations

In this analysis, we focus on the evaluation of SIM_E5, SIM_E432 and SIM_E432_EFMM to examine which emission cycle best matches the observed seasonal cycle in $\delta^{13}C$. The peaks and amplitude of the observations are calculated from 30-day moving averages of the detrended data because the variations in the observations are high even after the smooth-fitting (Fig. 4).

At the SPO station, the observations of $\Delta\delta^{13}C$ show a small enrichment in the early months of the year, after which a gradual depletion is observed until SH spring (NH autumn) (Fig. 4). From September to the end of the year, the observations show a gradual enrichment. The amplitude of the mean seasonal cycle over 2002–2012 is 0.15 ‰. The standard deviation of the





**Figure 4.** Detrended modelled and observed average seasonal cycles during 2002–2012 at Alert, Niwot Ridge, and South Pole. Gray dots are individual detrended observations. Note the differences of the y-axis scales.

detrended observations is $\sigma = 0.048$ ‰. In general, at the SPO station, the model captures the observed seasonal cycle of $\Delta CH_4$ and $\Delta\delta^{13}C$ well. Modelled $\Delta\delta^{13}C$ follows the shape of seasonal cycle well, but the modelled seasonal cycle amplitudes are smaller (0.13–0.14 ‰) compared to the observations (0.15 ‰). For $\Delta CH_4$, modelled amplitudes are 7.7–9.8 % larger than the observations, mainly due to a deeper minimum in the model. As expected, there are no major differences between the simulations in SPO as the site is far from the emissions sources.

At ALT, the observations of $\Delta\delta^{13}C$ show a gradual enrichment until summer (~DOY 200), followed by a strong depletion in autumn (until ~DOY = 260) (Fig. 4). After this depletion, the observations show a gradual enrichment towards the end of the year, continuing to the next summer. The seasonal amplitude in the observations is 0.45 ‰, and model estimates largely underestimate this amplitude by over 50 % (0.13–0.19 ‰). This is mainly because the model is not capturing the strong $\Delta\delta^{13}C$ depletion in summer and autumn (see Section 4 for discussions on reasons). The simulations SIM_E5 and SIM_E432_EFMMN show enrichment in $\Delta\delta^{13}C$ from the beginning of the year until DOY $\approx$ 200, similar to the observations. However, SIM_E432



shows depletion during DOY ~50 to 120. This suggests that the EFMM emission cycle in EDGAR v4.3.2 causes the depletion in spring, as was shown in the zonal mean estimates (see Section 3.1.1). In addition, the model reaches the $\Delta\delta^{13}$C maximum and minimum ~20 days later than the observations. Modelled $\Delta$CH$_4$ has smaller amplitudes (SIM_E5 31.6 ppb, SIM_E432 37.8 ppb, SIM_E432_EFMM 34.5 ppb) than the observations (50.1 ppb). $\Delta$CH$_4$ is smaller than observed for approx. DOY

< 100. Observed $\Delta$CH$_4$ reaches its maximum at approx. DOY = 50 in spring, but the modelled $\Delta$CH$_4$ maximum is 75 days later. The observed $\Delta$CH$_4$ reaches its minimum at approx. DOY = 200, but the modelled $\Delta$CH$_4$ at approx. DOY = 240. In addition, the increase in modelled $\Delta$CH$_4$ remains smaller compared to observations.The shape of the $\Delta$CH$_4$ cycle is closest to the observations in SIM_E5, while the amplitude is closest to SIM_E432. In general, when the modelled $\Delta\delta^{13}$C is lower than the observations, the modelled $\Delta$CH$_4$ is higher than the observations, except during approx. DOY = 25–100 when both

$\Delta\delta^{13}$C and $\Delta$CH$_4$ are lower than observations. The differences between the model estimates and observations may be due to smaller magnitude of wetland CH$_4$ emissions (higher values of $\delta^{13}$C and lower magnitude of CH$_4$) or smaller magnitude of OH sink (higher values of $\delta^{13}$C). However, increasing wetland CH$_4$ emissions in summer would cause larger discrepancies in CH$_4$ abundance in summer-autumn (Warwick et al., 2016), and therefore, the magnitude of wetland CH$_4$ emissions are probably not only the cause for the observed discrepancies between model and observations. In addition, higher OH concentrations during

spring and early summer, and lower OH concentrations in autumn could lead to a better match with the observations. Note that changes in the emissions affect the modelled CH$_4$ and $\delta^{13}$C with some lag (see Section 3.1.2), but changes in OH would lead to an instantaneous effect.

At NWR (Fig. 4) the $\Delta\delta^{13}$C observations increase gradually from the beginning of the year until summer (approx. DOY $\leq$ 200), and decreases until early winter (approx. DOY $\leq$ 300). The amplitude of the observed $\Delta\delta^{13}$C is 0.3 ‰, with $\sigma = 0.078$ ‰,

and the modelled amplitudes (0.12–0.13 ‰) are again less than half of the observations. All three simulations show depletion in summer, ~50 days later than observations suggest. The depletion in summer and autumn is not as strong as in ALT, and the models follow the depletion better than in ALT, although the minima are slightly shallower compared to the observations. Compared to ALT, the summer enrichment of $\Delta\delta^{13}$C is relatively stronger at NWR. The observations show strong enrichment at around DOY = 200, but none of the simulation could reproduce the peak. The discrepancies are high, especially in the

simulations using EDGAR v4.3.2. In SIM_E432_EFMMNS the modelled $\Delta\delta^{13}$C stays constant until around DOY = 125, while the modelled $\Delta\delta^{13}$C first increases in SIM_E5. SIM_E432 again shows a strong depletion between DOY 50–100, as in ALT. Nevertheless, all the simulations reach a similar maximum at around DOY = 225.

The amplitude of the modelled $\Delta$CH$_4$ is about 7 % smaller in SIM_E5 (31.5 ppb) and 18–31 % larger in SIM_E432 (44.0 ppb) and in SIM_E432_EFMMNS (39.8 ppb) compared to the observations (33.7 ppb). The modelled $\Delta$CH$_4$ follow

the observations well until DOY = 75 in SIM_E432_EFMMNS. The observations reach at maximum at around DOY = 75. After DOY = 75 that, modelled CH$_4$ in SIM_E432 and SIM_E432_EFMMNS continues to increase until approx. DOY = 125, while SIM_E5 remains more or less constant until DOY=125, following the observations well. The timing of the minimum, however, does not differ much: the observations reach a minimum value at approx. DOY = 200, while all the simulations reach its minima at about DOY = 225. All the simulations show slight underestimation of $\Delta$CH$_4$ in winter. The differences

between modelled and observed $\Delta\delta^{13}$C in SIM_E5 therefore could be due to wrong proportion of biogenic (heavily depleted)





to fossil based (less depleted) emissions source especially during summer. The differences in SIM_E432 in spring suggest that the biogenic EFMM emissions are probably overestimated. However, although the seasonal cycle of EFMM emissions is removed, the spring discrepancies remain (SIM_E432_EFMMNS). In addition, the differences in both $\Delta CH_4$ and $\Delta\delta^{13}C$ in winter suggest underestimation of biogenic emissions from all simulations.

### 455 3.3 Zonal means in the stratosphere

In this section, we examine the seasonal cycle in the zonal mean of $\Delta\delta^{13}C$ in the stratosphere, similarly to Section 3.1.1, averaged over the eight top most layers of TM5 (corresponds to levels with pressure between approx. 88–0 hPa). The stratospheric chemistry and transport are important such that those also affect the seasonal cycle in the troposphere to some extent. Note however, $\delta^{13}C$ observations in the upper atmosphere is limited to validate the modelled seasonal cycles, so this is simply a
summary of the model results. Nevertheless, the model is able to simulate realistic $\Delta CH_4$ and $\delta^{13}C$ profiles for the stratosphere compared to e.g. Röckmann et al. (2011) (Fig. not shown).

In the spin-up simulations, we could see that the seasonal cycle of $\delta^{13}C$ in the tropical stratosphere is influenced by the transport of air from the troposphere. The timescale for reaching a stable $\delta^{13}C$ seasonal cycle was therefore similar to that in the troposphere, i.e. ca. 2 years (Supplementary Fig. S5). Therefore, we consider the detrended seasonal cycle from 2002–
2012 for the analysis. The temperature inversion in the stable stratified stratospheric air leads to slow vertical transport, with transport timescales of a few years. The slow transport caused the seasonal cycle of $\delta^{13}C$ to stabilize in less time in mid and high latitudes than in the tropics (Figure not shown) in the spin-up simulations. Therefore, the timescale for chemistry is shorter than for transport, and the $\delta^{13}C$ is less influenced by the tropospheric air masses and the $\delta^{13}C$ corresponds to the KIE of the stratospheric sinks.

The photochemical sinks in the stratosphere together with the Brewer-Dobson circulation causes the amplitudes in the seasonal cycle of both $\Delta CH_4$ and $\Delta\delta^{13}C$ to become much larger in the stratosphere (47–103 ppb, and 0.53–1.28 ‰) than in the troposphere. The shape of the $\Delta\delta^{13}C$ seasonal cycles is again generally mirroring the cycle of $\Delta CH_4$. This is expected, as the KIE of all sinks (OH, Cl and O($^1$D)) in the stratosphere is larger than 1.

The phase shifts are less clear in the stratosphere (correlation $p_0 \leq 0.96$ and $R^2 \geq 0.92$), indicating that the effect of emissions
on $\Delta\delta^{13}$ seasonality is small (Supplementary Fig. S6). This is in line with the conclusion from the spin-up simulations described above. This also results in negligible differences in Delta$\delta^{13}$ seasonality between the simulations that differ in their surface emissions (figure not shown).

The phase ellipse at 60° S–90° S yet does not form a straight line, such that the increase in $\Delta\delta^{13}C$ in the beginning of the year is relatively slower than the decrease in $\Delta CH_4$ compared to the second half of the year (Supplementary Fig. S6). This is
probably due to seasonal effect of the transport as well as seasonal changes in the species that contribute to the sink. At high latitudes, the Brewer-Dobson circulation affects the methane distribution strongly, leading to a decrease in $CH_4$ towards winter with $^{13}CH_4$ enriched air masses. Because the Brewer-Dobson circulation is stronger in the Southern Hemisphere, the effect is especially prominent in the southern high latitudes (60° S–90° S) (Fig. 5 and Supplementary Fig. S6). The maxima in the tropical region (30° S–EQ and EQ–30° N) follow the position of the strongest vertical motion, i.e. influx of methane rich air





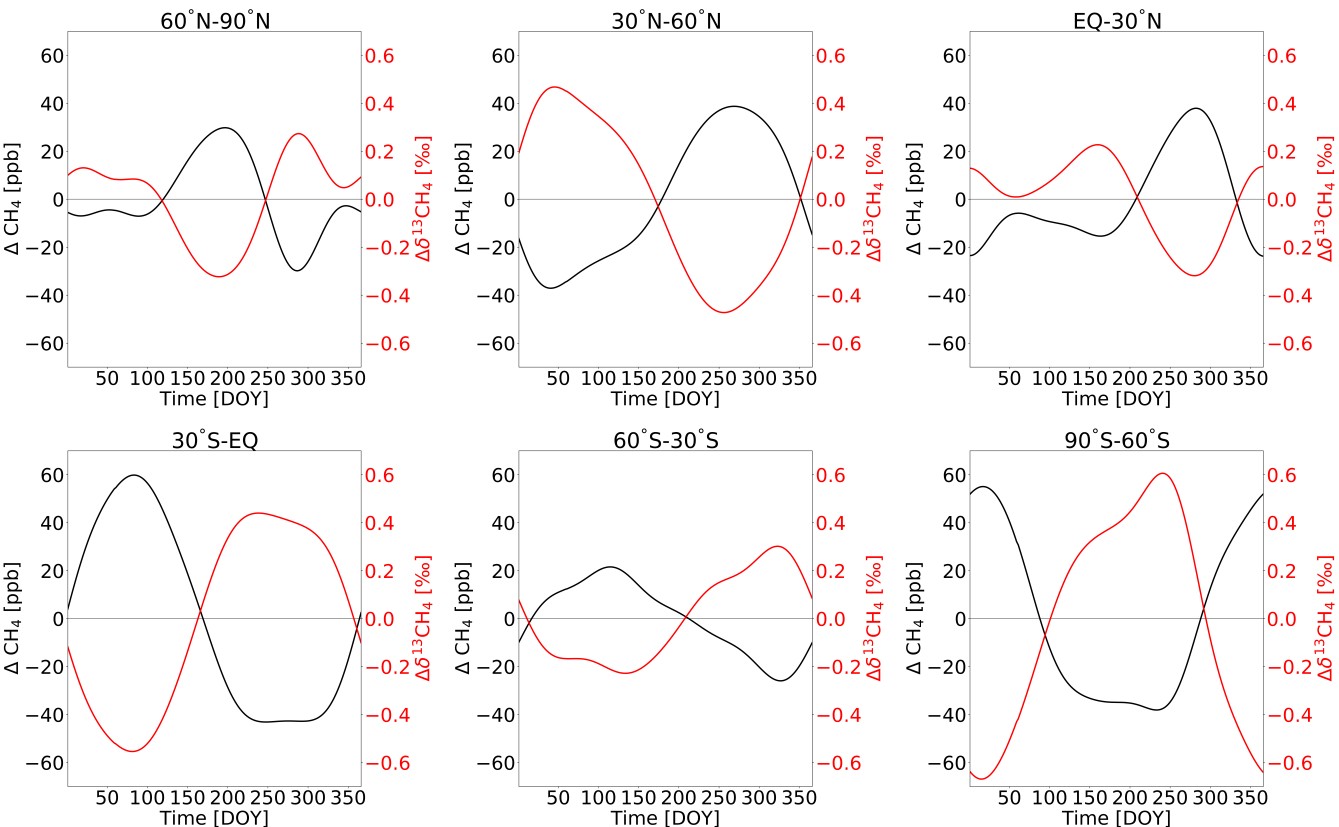

**Figure 5.** Seasonal cycle in zonal mean $\Delta CH_4$ and $\Delta\delta^{13}C$. The data is averaged over the eight uppermost model layers. The daily averages are calculated from 2002–2012 based on SIM_E5.

from the troposphere. The asymmetry in the amplitude in the tropical seasonal cycles show the difference in the strength of the vertical motion.

## 4 Discussion

### 4.1 Isotope signatures

The seasonality of $\delta^{13}C$ depends on by the isotopic signature of the emissions. Ganesan et al. (2018) showed that changing the source signatures by about ±13 ‰ affects the modelled $\delta^{13}C$ more than ± 0.5 ‰. Furthermore, our spin-up tests also indicate that changing the isotopic signature by ± 1 ‰ can reduce/increase the $\delta^{13}C$ seasonal cycle amplitude by 1.5–7 times, and the timing of minima/maxima by 7–130 days at 30° latitudinal bands (Supplementary Fig. S7). Considering that our results agree well with the observations at the South Pole, we could assume that the currently used isotopic signatures are correct in a broad





sense, and that more detailed and a better spatial and temporal distribution of the signature values would improve the agreement
with observations.

Although we have used a recently published spatial distributions of source signatures where available, there are still large uncertainties in the modelled $\delta^{13}$C values due to e.g. the vegetation types, especially tropical wetlands (Ganesan et al., 2018). Tropical wetlands are more enriched with C-13 compared to high-latitude wetlands partly because warmer wetlands have often a thicker oxic layer, and in part because of differences in methanogenic communities, and in part because of different plant
material; high-latitude wetlands precursor plant material is C3, which is depleted in C-13, while in the Tropics the dominant plant material is C4 (Fisher et al., 2011).

The coal source signatures vary depending on coal types, depths, coalification processes, type of mining and coal rank (Zazzeri et al., 2016), but only limited measurements and country-level data for coal mining types are available, and may be misreported (Feinberg et al., 2018) as the coal source signature can also vary within in a country (Zazzeri et al., 2016).
EFMM source signatures are dependent on types of feed (Feinberg et al., 2018; Chang et al., 2019; Levin et al., 1993), and therefore isotopic signature can be determined by the fraction of major vegetation types (Chang et al., 2019), i.e. C4 and C3 plants (Still et al., 2003). However, the vegetation distribution in a region may not correspond to the livestock diet. Ruminant C3 diet leads to more depleted values of $\delta^{13}$C emissions (Brownlow et al., 2017). Manure management isotopic source signatures depend on manure type; liquid manure is more depleted in $^{13}$CH$_4$, than manure pile emissions (Levin et al., 1993).
Rice cultivation methods also affect the source signatures. Mulching cultivation leads to higher values of $\delta^{13}$C compared to traditional cultivation, and to lower CH$_4$ total emissions (Zhang et al., 2017). In this study, we used a global uniform value of -62.1 ‰ for RICE emissions, but measured values vary between approximately -45 ‰ and -65 ‰ at three different rice fields in China (Zhang et al., 2017). The seasonality of $\delta^{13}$C is determined by cultivation method, tillage and Nitrogen (N) fertilization, but is also controlled by drainage. For instance, permanently flooded rice fields have enriched $\delta^{13}$C values at the beginning
of the crop season followed by a rapid depletion in $^{13}$CH$_4$ and towards end of the season $\delta^{13}$C values become enriched again (Zhang et al., 2017). In double harvest fields, the $\delta^{13}$CH$_4$ values are depleted immediately after drainage (Zhang et al., 2017).

Biomass burning emissions have strong $^{13}$CH$_4$ seasonality due to burning activity and vegetation types, especially in African Savannah, (Brownlow et al., 2017). C4 plants in African Savannah are abundant, and during the burning period, usually between December and March, $\delta^{13}$C is shifted towards less negative (Brownlow et al., 2017). Pine forest burns in southeastern USA are
more depleted with $^{13}$CH$_4$ (-21 ‰ to -29.5 ‰) compared to African grassland burning (-16.6 ‰ to -26.1 ‰), while African woodland burns produce methane with -30 ‰ (Chanton et al., 2000). The $\delta^{13}$CH$_4$ signature of biomass burning varies during different phases, i.e., the smouldering phase it produces more depleted methane compared to flaming phase (Chanton et al., 2000). However, the main factor determining $\delta^{13}$C is plant type C3 or C4 (Quay et al., 1991).

In addition, we acknowledge that KIE of soil sinks varies among soil types. Snover and Quay (2000) measured 1.0173 and
1.0181 for temperate grasslands and forest in Washington Sate, USA, respectively. In addition, Tyler et al. (1994b) measured a value of 1.022 in College Woods in Durham, New Hampshire, USA. Reeburgh et al. (1997) measured values between 1.022–1.025 in boreal forest in Bonanza Creek, Alaska, USA. Those may also vary with temperature and CH$_4$ concentration due to variation of biological KIE (Tyler et al., 1994b).





Furthermore, source signatures may have seasonal variations. These variations have been reported for biogenic sources,
such as wetlands, rice cultivation and ruminants. Wetlands are expected to have temporal variations depending on emission
processes (Fisher et al., 2017). Forested bogs have larger temporal variations of $\delta^{13}$C emissions compared to poor fen, and the
CH$_4$ emitted from fen is more enriched with C-13 compared to bog emissions (Kelly et al., 1992).

Rice cultivation source signatures also have temporal variations (Tyler et al., 1994a; Bergamaschi, 1997). The exact reasons
for the variations are unclear, but possible explanations include changes in the methanogensis pathway (Whiticar et al., 1986),
changes in the relative rates of production and oxidation of CH$_4$ with time (Kelly et al., 1992) and a temperature dependence
of the isotope effects in the production of methane (Blair et al., 1993).

As ruminant source signatures are dependent on feed type, the source signatures may have a seasonal cycle and annual
variations when the cattle diet changes (Lopez et al., 2017). Seasonal changes in manure management may be affected similarly,
but the exact values are yet not well quantified.

## 540  4.2   Seasonal cycle of CH$_4$ emissions

The modelled $\delta^{13}$C cycle is found to be mostly affected by wetland emissions, and therefore, $\delta^{13}$C measurements could be used
to evaluate CH$_4$ emission magnitude and seasonal cycle of wetland emissions. Wetland CH$_4$ emissions at high latitudes (north
of 50° N) estimated by process-based models (Melton et al., 2013; Bloom et al., 2017) find a maximum in May-August, and
inverse models (Bousquet et al., 2011) find a maximum in July. Studies based on $\delta^{13}$C measurements indicate that the high-
latitude CH$_4$ emissions likely peak in August. (Warwick et al., 2016; Fujita et al., 2018; Thompson et al., 2018). Thompson
et al. (2017) showed that the LPJ-DGVM vegetation model estimates CH$_4$ wetland emissions to peak between May and June,
depending on the region within the Northern High Latitudes (NHL; north of 50° N), which is considerably earlier than inversion
models (July–September). LPX-Bern v1.4 wetland emissions used in this study has maximum emissions at NHL occurring in
September, later than other studies. Aalto et al. (2021) showed that the main reasons for the late maximum in LPX-Bern v1.4 is
the strong precipitation dependence of wetland emissions. In addition, the maximum NHL CH$_4$ emission in LPX-Bern v1.4 is
approximately 3 Tg CH$_4$ month$^{-1}$ lower than those in Fung et al. (1991), used in Warwick et al. (2016), and the Saunois et al.
(2020) bottom-up model ensemble mean. Tenkanen et al. (2021) also showed that emissions in the NHL are ~6 Tg CH$_4$/yr
greater than in LPX-Bern v1.4, based on an inversion using ground-based atmospheric CH$_4$ data. Top-down and bottom-up
estimates differ much in 60° N–90° N, with top-down estimates being 40–49 % of the bottom-up estimates (Saunois et al.,
2020). However, Warwick et al. (2016) showed that the emission magnitude should be approximately doubled to resolve the
$\delta^{13}$C amplitude as observed in e.g. ALT. It may be unrealistic to assume that wetland emissions in high northern latitudes are
underestimated so significantly, but spatial distribution of the emissions (reference), the other natural biogenic emissions, such
as those from inland water systems (Rosentreter et al., 2021) and the effects of the upland soil sink on $^{13}$C (Oh et al., 2020)
may well be reasons for the underestimation of the $\delta^{13}$C amplitude at high latitudes in this study.

In the Amazon, the climatological monthly biogenic CH$_4$ flux based on column budgeting (difference of the CH$_4$ column
content is due to the sum of fluxes along the air parcel path) displays two peaks, one in February and another in September–
October (Basso et al., 2016). An inversion study over Brazil suggests wetland emissions peak in February and March (Tun-





nicliffe et al., 2020). Our 30° S–EQ averaged wetland emission also shows a high peak in February. However, the peak in September–October is not present. However, results from SCHIAMACHY (Bloom et al., 2012) over the Amazon show that

in SH Tropics, the peak is likely to appear in December–February and near the equator the peak is in February–April and a bit to the north in June-August. These differences in wetland emission timing between NH and SH Tropics correspond to the LPX-Bern results used in this study. In addition, the ENSO phase also affects the inter-annual variability of wetland emissions in tropical wetlands, enhancing tropical wetland emission during La Niña (Pandey et al., 2017; Zhu et al., 2017).

Other important emissions in the Tropics is from the fires. Fires in the Amazonian region occur in August-December, with

a peak in November (Basso et al., 2016), in line with the GFED emissions used in this study.

$CH_4$ emissions from rice follow the rice growing calendar (Cao et al., 1996). Cao et al. (1996) modelled $CH_4$ emission from rice with a maximum in July–September north of 20° N, and in December–February in the south of 10° S, while near the equatorial regions emissions are high throughout the year, peaking in August. Zhang et al. (2016) also estimated global rice $CH_4$ emissions to peak in July–August. Measurements performed during the growing season (Bergamaschi, 1997; Tyler et al.,

1994a) agree with these estimates. $CH_4$ emissions from rice cultivation, provided by EDGAR v5.0 correspond better to these estimates and numbers than emissions from EDGAR v4.3.2.

The seasonal cycle of $CH_4$ emissions from the EFMM sector also varies considerably between the EDGAR versions. A study based on dairy cows and ewes in New-Zealand showed that the seasonal changes follow the different amounts of feed intake and seasonal variation in milk production for dairy cows compared to ewes (Ulyatt et al., 2002). The study found maximum

emissions in September, while the estimated minimum emissions vary between June and March, depending on whether feed intake is taken into account. This is more in line with EDGAR v4.3.2 compared to v5.0 at 30° S–90° S. A study performed in South Africa, concluded that Nguni and Boran cows produce more $CH_4$ per kilogram live weight of cow during the dry season (mid-May to October) compared to the wet season (November to early May) (Mapfumo et al., 2018), which partly corresponds to EFMM emissions peaking in August-October in EDGARv4.3.2 at 30° S–90° S. In contrast, Arndt et al. (2018)

measured emissions from Jersey cattle (primary breed) in California and found that animal housing (enteric fermentation as major source) had no seasonality, which corresponds well to EDGAR v5.0. Chen et al. (2018) found only weak seasonality for livestock emissions, but warmer air temperature is likely to enhance manure $CH_4$ emissions, contradicting both EDGAR versions.

Manure management has a seasonal cycle depending on liquid manure storage empty-full level and the temperature of the

manure (VanderZaag et al., 2014). VanderZaag et al. (2014) measured whole farm emissions (EFMM) in Ontario, Canada to be larger in autumn compared to spring, contrary to EDGAR v4.3.2 peak at 30° N–60° N, which is opposite. Cárdenas et al. (2021) showed that the temperature at which manure is stored affects the amount of emitted $CH_4$, such that during summer CH4 emission is greater, but when the temperature drops below 13.39 °C, $CH_4$ emissions from manure are low. Arndt et al. (2018) also concluded that $CH_4$ emission measured in California, USA from liquid manure storage are larger during summer

than in the winter. Husted (1994) also found that $CH_4$ emission from slurries increases with storage temperatures. However, these findings do not correspond to either EDGAR version at 30° N–60° N.





### 4.3 Atmospheric sinks

OH is the largest $CH_4$ sink in the atmosphere, and it removes $^{12}CH_4$ faster than $^{13}CH_4$. The seasonal cycle of OH is affected by humidity, clouds, temperature and forest fires, and Rohrer and Berresheim (2006) estimated that ~23 % of long-term variations in OH concentrations can be explained by seasonality. Lowe et al. (2004) speculated that their underestimation of the $\delta^{13}C$ seasonal cycle amplitude in the Tropics may be associated with the OH sink, while Allan et al. (2001b) suggested the overestimation of $CH_4$ seasonal cycle in the model is associated with an overestimate of OH sink by more than 28 %. We examined the $\delta^{13}C$ seasonal cycle by changing the seasonal cycle amplitude of the OH concentrations by ± 10 %, but the effect on the tropospheric $\delta^{13}C$ seasonal cycle with respect to $CH_4$ was insignificant even in the Tropics (figure not shown).

Marine boundary layer Cl is thought to have a non-negligible contribution in $CH_4$ sink, with estimates 5–25 Tg yr$^{-1}$ in range (Allan et al., 2001b, 2007). Due to its stronger fractionation compared to that of OH, underestimation of tropospheric Cl will likely lead to underestimation of the $\delta^{13}C$ seasonal cycle amplitude in the troposphere, assuming that Cl concentration has similar seasonality as OH (Allan et al., 2007; Wang et al., 2019). In this study, we did not include the tropospheric Cl sink, but could nevertheless reproduce the $CH_4$ and $\delta^{13}C$ seasonal cycle measured at the South Pole reasonably well. As emission seasonality affects this site only marginally, the seasonality at the SPO is mostly driven by atmospheric sinks. Although it has been shown that marine BL Cl concentration is the highest in the Tropics (e.g. Hossaini et al., 2016), our results support the recent study by Gromov et al. (2018) who suggested that the contribution of the tropospheric Cl sink to atmospheric $CH_4$ budgets are small. Our model results show that the $CH_4$ to $\delta^{13}C$ ratio does not follow the theoretical KIE line at the southern high latitude, similarly to the observation based study by Allan et al. (2001b). They argued that the kinetic isotope fractionation at a site in the SH extratropics require an $CH_4$ oxidation pathway by Cl.

### 5 Summary and conclusions

We performed a global analysis of how different $CH_4$ emission sources influence the $\delta^{13}C$ seasonal cycle, averaged over 30° latitudinal bands during 2000–2012, using the TM5 atmospheric chemistry transport model. Based on the simulation results, wetland emissions are found to be the dominant $CH_4$ source driving the $\delta^{13}C$ seasonal cycle, apart from atmospheric sinks. Wetland emissions are the key, especially in determining the timing of $\delta^{13}C$ minimum peaks in the NH. Comparisons using EDGAR v5.0, v4.3.2, and additional emission scenarios removing the seasonal cycle of emissions from enteric fermentation and manure management (EFMM) in v4.3.2 show that EFMM emissions in v4.3.2 are responsible for the unique $\delta^{13}C$ depletion in the NH in spring.

Seasonal cycles in $\Delta\delta^{13}C$ are reverse of $\Delta CH_4$ cycles in general, with a significant anti-correlation. However, due to the effects of the emissions, the phase ellipses do not form a straight line, but rather oval shapes, especially in the NH. At 30° S–EQ, the phase ellipse forms an irregular shape due to the effect of wetland and biomass burning emissions, which have distinct isotopic signatures and emission cycles. We also found that the effect of sinks other than OH contribute to the $\Delta\delta^{13}C$ cycle in relation to $\Delta CH_4$ cycle. The phase ellipse did not become a straight line in 60° N–90° N even when seasonality of the



emissions is removed, suggesting the effect of soil sink, and the KIE-line slope in the deep SH (90° S–60° S) did not exactly

follow those when only the OH sink is considered.

Compared to INSTAAR observations at the South Pole station, Antarctica, the model is able to reproduce the $\Delta\delta^{13}$C seasonal cycle well, indicating that seasonality of the sinks and to a lesser extend in the emissions in the model are at the right level. For $\delta^{13}$C observations closer to the emission sources, at Alert, Canada and Niwot Ridge, USA, the moel estimates using EDGAR v5.0 are in closer agreement to the observations than those using v4.3.2. The seasonal cycle of EFMM causes a $\delta^{13}$C depletion

in spring in Alert and Niwot Ridge, which is not in the observations, suggesting that the seasonal cycle of EFMM is not correct in EDGAR v4.3.2. In addition, the modelled $\Delta\delta^{13}$C seasonal cycle amplitude is underestimated and maximum and minimum for $\Delta\delta^{13}$C are around 20 days later than the observations in Alert. The cause of these discrepancies may be underestimation of wetland $CH_4$ emissions in the northern high latitudes in summer, although some other factors, e.g. timing of wetland emission peak, seasonal cycle of OH, and isotopic signatures could also affect the simulated seasonal cycles.

Seasonal cycle amplitudes of both $\Delta\delta^{13}$C and $\Delta CH_4$ are much larger in the stratosphere than in the troposphere. The effect of emissions is negligible in the stratosphere, and the $\delta^{13}$C seasonality is therefore driven by the atmospheric sinks. However, due to lack of $\delta^{13}$C observations at high altitudes, it is difficult to evaluate whether our estimated seasonal cycles are realistic.

Here, we have focused on the effects of emissions and their source signatures on the simulated seasonal cycle of $\delta^{13}$C. There is an increasing number of studies examining the spatial and temporal distributions of emission signatures, but further research

at regional to global scales is needed to examine global changes of $\delta^{13}$C. Furthermore, a step forward to better understand the different source contributions would be to build an atmospheric inversion, and will be the scope of our next study.

*Code and data availability.* The source code of TM5 used in this paper is available from https://doi.org/10.23729/966bb3fa-6c15-43d d-94d2-e80c5f7ce2f7. The data presented will be provided on request from the corresponding author. The atmospheric $CH_4$ and $\delta^{13}$C measurements are available from NOAA/GML data server https://www.esrl.noaa.gov/gmd/dv/data/

*Author contributions.* VK, AT and TA designed the experiments. VK and AT developed the model code with help from LB, SH, AS and MK. VK and AT performed the simulations. VK, AT, LB, PM and TA performed the analysis. ED, SM and JW provided the observational data. VK prepared the manuscript with contributions from all co-authors.

*Acknowledgements.* We would like to thank Magnus Ehrnrooth Foundation, the Vilho, Yrjö and Kalle Väisälä Foundation, Academy of Finland (307331 UPFORMET), EU-H2020 VERIFY and ESA-MethEO for financial support. The VERIFY project has received funding

from the European Union's Horizon 2020 research and innovation programme under grant agreement No. 776810. Maarten Krol is supported by funding from the European Research Council (ERC) under the European Union's Horizon 2020 research and innovation programme under grant agreement No 742798. We thank Xin Lan for the valuable discussion and Aryeh Feinberg for sharing information that greatly assisted this work. We also thank Fortunat Joos, Sebastian Lienert and Jurek Müller for providing LPX-Bern v1.4 data and helping to use it.



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
