# Peer review of "Role of emission sources and atmospheric sink on the seasonal cycle of CH4 and $\delta^{13}$ -CH4: analysis based on the atmospheric chemistry transport model TM5"

_Atmospheric Chemistry and Physics, 2021_

## Referee Comment (RC1)

The study of Kangasaho et al. investigates the influence of emission sources and atmospheric sinks of CH4 on the seasonal cycle of CH4 and d13C based on simulations with the atmospheric transport model TM5.

Similar studies, which laid the foundations to our understanding of the seasonal cycles of CH4 and d13C and their phase relations, have already been published in the early 2000s. Bergamaschi et al. (2000), for example, interpreted measurements at Izaña using global simulations with the TM2 model. Allan et al. (2001) also performed TM2 simulations and focused on the interpretation of the seasonal cycles in the extratropical southern hemisphere. Together they established a theoretical understanding of the phase relations between the seasonal cycles of CH4 and d13C and of phase ellipses also used here. In those studies, CH4 and d13C were simulated separately for different emission sources and then added up to the totals, which enabled a thorough analysis and understanding of the results. In the study of Bergamaschi et al. (2000), this information was further used in a simple inversion framework to adjust the emission sources to better match the CH4 and d13C observations at Izaña.

In the study of Kangasaho et al., instead, only two separate tracers were simulated (total CH4 and total d13C), while the influence of different emission sources or different seasonal emission cycles was analyzed through a set of sensitivity runs (see Table 2). This setup seems much less flexible and has clear limitations with respect to the interpretability of the results.

It is difficult to see what we can learn from this study that we didn't know already before. The main goal seems to be to reassess the seasonal cycles given our present best knowledge of sources and sinks and their isotopic signatures and to evaluate whether the seasonal emission cycles proposed in the EDGAR emission inventories are consistent with observed seasonal cycles in CH4 and d13C (and their phase relations).

However, the goals of the paper are not clearly stated and the chosen approaches are rather poorly motivated. The paper provides mainly technical descriptions of the model setup and simulations but fails to motivate the design choices. Another major weakness is a lengthy and unfocused presentation and discussion of the results with long descriptions of what is seen in figures but failing to extract the essential information in a concise manner.

Thus, I cannot recommend publication of the manuscript in its present form. It either needs

- a much clearer explanation of the goals of the study
- a much clearer motivation of the simulation setup and explanation of the methods used to analyze the results
- a much shorter and more concise presentation and discussion of the results. The present results section is unnecessarily lengthy and unfocused (see also points below)

Another, probably better option, would be to redesign the simulation experiment, notably by not only simulating two separate tracers but simulating CH4 and d13C from the different sources individually as in those earlier studies. Such an experiment could also include tracers with seasonally varying and seasonally constant emissions.

**Further major points:**

Section 3.1.1 is very hard to follow and in fact painful to read. The text basically describes what is seen in the figure but adds a lot of numbers that often seem quite irrelevant and certainly do not help telling a concise story. If needed, such numbers could be summarized in a table (latitude band, amplitude of seasonal cycle, month of maximum, month of minimum, etc.), but actually I don't see the point in presenting all these details at all. This section could easily be cut to one third by only focusing on the main points and leaving out all the fine details. There is also a lot of speculation in this section ("we suspect that", "this indicates that", "this could explain" etc.) which is related to the weaknesses of the simulation design mentioned earlier.

The main take home messages from Figure 2 could simply be: (i) the seasonal cycles of CH4 and d13C are to first order in antiphase, i.e. maxima in CH4 correspond to minima in d13C and vice versa (consistent with what we know from previous studies). Minima (maxima) in d13C, however, tend to be xx months earlier than the maxima (minima) in CH4. (ii) at high latitudes in the southern hemisphere, where emissions are small and the seasonal cycle is primarily determined by sinks, differences between the different sensitivity simulations are very small. (iii) At northern mid- to high latitudes, differences in the seasonal timing of emissions has a pronounced influence on the seasonal cycle of CH4 and especially of d13C. In the simulation with constant emissions (E5_WETNS), for example, wetland emissions are higher in winter to spring and lower in summer to autumn compared to the reference simulation (E5). This leads to higher CH4 north of 30°N in spring and lower in autumn and, because of the strongly depleted isotopic signature, to a minimum in d13C in early spring (not seen in E5) and a maximum in autumn which is delayed compared to E5 by xx months. (iv) At low southern latitudes (0-30°S), there is a double peak structure in CH4 due to xyz and, in case of constant emissions, also in d13C. When seasonal variations in emissions is accounted for (ref. simulation E5), however, the double-peak in d13C disappears because d13C is pulled down by strong wetland emissions in the southern hemisphere summer and autumn and pushed up by strong biomass burning emissions in the southern hemisphere spring. A few further details may be added, but further discussions could be presented in the context of the comparison with the observations in Section 3.2.

Similarly, the presentation of the results in terms of phase ellipsoids needs to be cut strongly and could be presented along the same line of arguments as above.

One of the conclusions drawn in the study is that the seasonal cycle of wetland emissions is the dominant driver of the seasonal cycle of d13C in the NH and tropics (see corresponding sentence in the abstract). However, this seems to be primarily based on the comparison between simulations E5_NS and E5_WETNS. Comparing simulations E432 and E432_EFMMNS would tell a different story, namely that a seasonal cycle in agricultural CH4 emissions could also perturb the d13C seasonal cycle. Which effect is closer to reality should be decided based on the comparison against observations and not just on the comparison between individual sensitivity runs. Note that in all simulations using EDGAR 5 (E5, E5_WETNS, E5_NS) the agriculture sector has no seasonal variability, too, and therefore it is clear that this sector cannot contribute to differences between these runs. The conclusion drawn on page 12, line

280 (and again in the abstract and conclusions section), is therefore not valid. The main difference between simulations E5_WETNS and E5_NS (at least in the southern hemisphere) is that in E5_WETNS biomass burning emissions have a seasonal cycle but those of E5_NS have not. Comparing these two simulations tells a lot about the influence of biomass burning emissions (see e.g. panel for latitude band 0-30°S in Fig. 2), but this is hardly addressed in the paper.

Section 3.3 describes seasonal cycles in the stratosphere. This section has several weaknesses. First, it is poorly connected to the rest of the paper focusing on the troposphere. How stratosphere-troposphere-exchange affects the seasonal cycles of CH4 and d13C would actually be an interesting topic, but the simulation setup does not allow addressing this question. Second, averaging the seasonal cycles over a large vertical range from 88 hPa to the top of the atmosphere does not seem a good idea, as there are probably significant differences in seasonality across this altitude range. The authors should rather focus on a single level (e.g. 80 hPa) or present vertical cross-sections/vertical profiles of the evolution in the stratosphere. Third, the discussion of the results is not convincing. For example, the authors argue that the strong amplitude of the seasonal cycles at the South Pole compared to the North Pole is due to the Brewer-Dobson circulation being stronger in the Southern Hemisphere, but actually the Brewer-Dobson circulation is stronger in the Northern Hemisphere, see e.g. Rosenlof (1995; https://agupubs.onlinelibrary.wiley.com/doi/10.1029/94JD03122).

Section 4.1 needs to be rewritten and shortened as well. Rather than discussing the results in the context of other studies, it basically presents a review of existing literature. This is not what I expect from a discussion section. Section 4.2 is slightly better but could also be shortened.

**Minor points:**

The concept of phase ellipsoids should be better explained in the introduction section or somewhere else in the paper. Without this it is difficult to follow the discussion in section 3.1.2. Why do sinks produce a line but emissions an ellipsoid? Actually, to my knowledge only seasonally changing emissions produce an ellipsoid.

Sections 2.2.1 and 2.2.2 presenting the anthropogenic and natural emissions are quite long. A lot of the information could be put in a table (category, data source, total emissions per category, amplitude of seasonal cycle, etc.) so that the text could be shortened by referring to the numbers in the table rather than listing all these numbers in the text.

What is the basis for the selection of the three observation sites ALT, NWR and SPO? Data availability? Distance from sources? Would there be other stations that could have been looked at? Why were they not considered?

Figure 2: It would be good to highlight the line corresponding to SIM_E5 as a thick black line as a reference. The discussion in Section 3.1.1 should mainly refer to this line and discuss other simulations with respect to their deviations from this reference.

**Small points, typos and grammar:**

The article needs substantial editing for grammar:

For example, wrong prepositions are used at many places, e.g. "in 60-90°N" instead of "AT 60-90°N" or "the effect in the d13C cycle" instead of "the effect ON the d13C cycle". The suggested corrections to the abstract below give an indication of the magnitude of this problem. I do not provide any corrections related to grammar for the rest of the manuscript.

Affiliations: Shouldn't it be INSTAAR rather than NSTAAR?

Abstract:

Page 1, Line 3: What do you mean by "examined"? This is a very vague term.

P1, L3: "Those includes" -> "Those include"

P1, L5: "in addition to other sources" -> in addition to other natural sources"

P1, L6: "global uniform value" -> "globally uniform value"

P1, L9: "in north of" -> "north of"

P1, L11: "timing of d13C seasonal minimum" -> "timing of the d13C seasonal minimum"

P1, L11, "days in 60°" -> "days at 60°"

P1, L14-15: "In light of this research, comparison to the observation" -> "These comparisons"

P1, L19: In what sense should the "proportion of biogenic to fossil based emissions" be revised? Please be more specific.

P2, L32: "much studied" -> "studied intensively"

P2, L36: Why should these emission cycles depend on "political decisions"?

P2, L52: "also warned that" -> "also cautioned that"

P3, L58: Sinks do not have a strong seasonality due to the KIE but simply due to the seasonality of OH (or Cl) radicals.

P3, L63: The d13C maximum was two months EARLIER not later than the CH4 minimum in those studies.

P3, L79: "that are a coarsening from" -> "corresponding to a subset of"

P3, L79: ", and vertical mixing was calculated" -> ". Convective vertical mixing was calculated"

P4, L86: reactions of CH4 with OH, Cl and O1D are chemical but not photochemical reactions.

P4, L90: It is not quite clear from the description whether the levels of Cl and O1D are prescribed in the stratosphere or whether the reaction rates are prescribed (which is not exactly the same). Please clarify.

P4, L117: I wouldn't say that the values in the stratosphere are "not important", but they are certainly "less important". Troposphere and stratosphere are linked through stratosphere-troposphere-exchange.

P5, L132: "enriched 13CH4" -> "enriched in 13CH4". The seasonal cycle "is" dominated, not "are" dominated.

P5, L135: What do you mean by "keeping the same annual totals"? The same as in 2010?

P5, L143: "due to the differences" -> "due to the following differences"

P5, L143-146: These sentences are a bit confusing and could probably be formulated more elegantly.

P5, L148: Delete "compared to v5.0".

P5, L150: "increasing trend" -> "increasing emissions"

P7, L183: For some sources, spatially varying isotopic signatures were used following Feinberg et al. (2018). It would be good to know in more detail how Feinberg et al. collected this information. Certainly, it was not generated "based on global chemistry-climate simulations with SOCOL".

Table 2: Replace "vary globally" by "varying globally" in the table caption.

P8, L184: Geological and wetland emissions should not be described together in a single sentence, because these are completely independent sources.

P8, L199: What are the values proposed by Monteil and how do they differ from those of Thompson? Why do you not list the values of Monteil also in Table 1?

P9,L202: It should probably be "In contrast" rather than "In addition".

P9, L203: I don't understand the statement "decimal values instead of rounded integers" at all (I can only guess what this means).

P9, L212: Why "In contrast"? NWR is also far away from sources and measures background air like SPO. I don't see the contrast.

P9, L213: "Rockies" -> "Rocky Mountains"

P10, L227: "we examined the seasonal cycle" -> "we examined the impact of the seasonal cycle"

P10, L252: "in 90°S – 60°S approximately" -> "at 90°S-60°S is approximately"

P12, L269: "in spring" -> "from spring"

P12, L274: "two maxima peaks" -> "two maxima"

P12, L275: "In south of" -> "South of"

P12, L277: "closer follows" -> "more closely follows"

P12, L282: shouldn't it be "increased" rather than "decreased"?

P12, L296: "only one month" -> "only during one month"

P13, L305: "the strong effect" -> "the strong contribution"

P13, L308: "and magnitude of autumn" -> "and the magnitude of the autumn"

P13, L318: "against that of" -> "against those of"

P15, L342: On page 4 the KIE of the reaction with OH had a value of 1.004. Here it has a value of -3.98 per mil. How do these numbers relate?

P17, L408: I can't see in Figure 4 that the modelled amplitude is larger "mainly due to a deeper minimum". The maximum seems to be equally overestimated.

P18, L429: "not only the cause" -> "not the only cause"

P18, L437: "depletion better than in ALT" -> "depletion better"

P18, L445: "reach at maximum" -> "reach a maximum"

P18, L446: "After DOY=75 that" -> "After DOY=75"

P18, L449: "reach its maximum" -> "reach their maximum"

P19, L466-467: I don't understand why the first sentence leads to the conclusion presented in the second sentence. It should rather be the other way round: "Because" chemistry time scale is shorter than transport d13C stabilized more quickly at mid- to high latitudes and not "therefore".

P19, L476: Remove the "Delta" and add a "C" after d13.

P20, L491: This sentence needs clarification. A factor 1.5 to 7 change in the amplitude of the seasonal cycle of d13C due to a simple change of the isotopic signature (of what??) by 1 per mil sounds like a huge effect, which could make any model simulations useless.

P21, L504: "within in a country" -> "within a country"

P22, L567: Here a "(reference)" is missing.

P24: L614-615: The last two sentences in this section contradict the previous sentence concluding that the chlorine sink is not important.

P24, L624: "are reverse of deltaCH4 cycles" -> "are opposite to those of deltaCH4"

P25, L633: "moel" -> "model"

P25, L633: "extend" -> "extent"

---

## Author Comment (AC1)

**Authors' response to anonymous referee #1**

In the following, *referee's comments are in italic*, authors' responses in normal font.

Please note that some of the Figure and Table numbers have changed due to modification.

Article:

Table 1 *new

Table 1 -> Table 2

Table 2 -> Table 3

Table 4 *new

Figure 5 *removed

Supplementary:

Table 3 * removed

Table 3 * new

Table 4 *new

Figure 4-> Figure 9

Figure 4 *new

Figure 5 *new

Figure 6 *new

Figure 7 *new

Figure 8 *new

Figure 5 -> Figure 10

Figure 6 *removed

Figure 7 *removed

*The study of Kangasaho et al. investigates the influence of emission sources and atmospheric sinks of CH4 on the seasonal cycle of CH4 and d13C based on simulations with the atmospheric transport model TM5.*

*Similar studies, which laid the foundations to our understanding of the seasonal cycles of CH4 and d13C and their phase relations, have already been published in the early 2000s. Bergamaschi et al.(2000), for example, interpreted measurements at Izaña using global simulations with the TM2 model. Allan et al. (2001) also performed TM2 simulations and focused on the interpretation of the seasonal cycles in the extratropical southern hemisphere. Together they established a theoretical understanding of the phase relations between the seasonal cycles of CH4 and d13C and of phase ellipses also used here. In those studies, CH4 and d13C were simulated separately for different emission sources and then added up to the totals, which enabled a thorough analysis and understanding of the results. In the study of Bergamaschi et al. (2000), this information was further used in a simple inversion framework to adjust the emission sources to better match the CH4 and d13C observations at Izaña.*

*In the study of Kangasaho et al., instead, only two separate tracers were simulated (total CH4 and total d13C), while the influence of different emission sources or different seasonal emission cycles was analyzed through a set of sensitivity runs (see Table 2). This setup seems much less flexible and has clear limitations with respect to the interpretability of the results.*

*It is difficult to see what we can learn from this study that we didn't know already before. The main goal seems to be to reassess the seasonal cycles given our present best knowledge of sources and sinks and their isotopic signatures and to evaluate whether the seasonal emission cycles proposed in the EDGAR emission inventories are consistent with observed seasonal cycles in CH4 and d13C (and their phase relations).*

*However, the goals of the paper are not clearly stated and the chosen approaches are rather poorly motivated. The paper provides mainly technical descriptions of the model setup and simulations but fails to motivate the design choices. Another major weakness is a lengthy and unfocused presentation and discussion of the results with long descriptions of what is seen in figures but failing to extract the essential information in a concise manner.*

We thank the anonymous referee #1 for providing the valuable feedback. We apologize that the current form of the paper did not meet expectations. We have now revised the manuscript intensively, following the reviewer's comments.

*Thus, I cannot recommend publication of the manuscript in its present form. It either needs*

*- a much clearer explanation of the goals of the study*

*- a much clearer motivation of the simulation setup and explanation of the methods used to analyze the results*

*- a much shorter and more concise presentation and discussion of the results. The present results section is unnecessarily lengthy and unfocused (see also points below)*

*Another, probably better option, would be to redesign the simulation experiment, notably by not only simulating two separate tracers but simulating CH4 and d13C from the different sources individually as in those earlier studies. Such an experiment could also include tracers with seasonally varying and seasonally constant emissions.*

We apologize for the misunderstandings and inconsistencies that arose because of the weak formulation that existed in the manuscript. In this revision, we have phrased our text more carefully, and had the full paper language edited by a native English speaker. We have now revised the goals of the study much clearer and explained our motivation to use these specific methods in the introduction section Lines (71-89).

The previous studies are old and therefore this topic is important to revisit. Compared to the previous studies, there is now available new observations from multiple stations and added information regarding the source specific isotopic signatures (including spatial variations). In addition, the chemistry atmospheric model has improved significantly from the previous studies and there is information available on the seasonality of anthropogenic sources.

We have rewritten the presentation and discussion of the results following the comments you provided in your feedback. However, we did not redesign the simulation experiments because this study was meant to be the basis for a follow-up inversion study. Simulating all different sources simultaneously would be unnecessary time consuming and both heavy and computationally expensive for this study.

We believe that the findings of this study are valuable themselves, as they will contribute to our knowledge of the seasonality of different emission categories, and we also evaluate the possible flaws in the seasonal cycle of anthropogenic emissions presented by the two recent EDGAR inventories v4.3.2 and v5.0. Furthermore, comparison to observations helps to identify which EDGAR version is more realistic. This study is designed to increase our understanding of $\delta^{13}$C distributions and how different CH$_4$ sources contribute to the seasonal cycle of $\delta^{13}$C.

We hope that the newly rephrased explanation of the goals of this study will now meet expectations and have a much clearer vision about the purpose of this study and what we can learn from this study.

**Further major points:**

*Section 3.1.1 is very hard to follow and in fact painful to read. The text basically describes what is seen in the figure but adds a lot of numbers that often seem quite irrelevant and certainly do not help telling a concise story. If needed, such numbers could be summarized in a table (latitude band, amplitude of seasonal cycle, month of maximum, month of minimum, etc.), but actually I don't see the point in presenting all these details at all. This section could easily be cut to one third by only focusing on the main points and leaving out all the fine details. There is also a lot of speculation in this section ("we suspect that", "this indicates that", "this could explain" etc.) which is related to the weaknesses of the simulation design mentioned earlier.*

*The main take home messages from Figure 2 could simply be: (i) the seasonal cycles of CH4 and d13C are to first order in antiphase, i.e. maxima in CH4 correspond to minima in d13C and vice versa (consistent with what we know from previous studies). Minima (maxima) in d13C, however, tend to be xx months*

*earlier than the maxima (minima) in CH4. (ii) at high latitudes in the southern hemisphere, where emissions are small and the seasonal cycle is primarily determined by sinks, differences between the different sensitivity simulations are very small. (iii) At northern mid- to high latitudes, differences in the seasonal timing of emissions has a pronounced influence on the seasonal cycle of CH4 and especially of d13C. In the simulation with constant emissions (E5_WETNS), for example, wetland emissions are higher in winter to spring and lower in summer to autumn compared to the reference simulation (E5). This leads to higher CH4 north of 30°N in spring and lower in autumn and, because of the strongly depleted isotopic signature, to a minimum in d13C in early spring (not seen in E5) and a maximum in autumn which is delayed compared to E5 by xx months. (iv) At low southern latitudes (0-30°S), there is a double peak structure in CH4 due to xyz and, in case of constant emissions, also in d13C. When seasonal variations in emissions is accounted for (ref. simulation E5), however, the double-peak in d13C disappears because d13C is pulled down by strong wetland emissions in the southern hemisphere summer and autumn and pushed up by strong biomass burning emissions in the southern hemisphere spring. A few further details may be added, but further discussions could be presented in the context of the comparison with the observations in Section 3.2. Similarly, the presentation of the results in terms of phase ellipsoids needs to be cut strongly and could be presented along the same line of arguments as above.*

We apologize for the unnecessary excessive information presented in these sections. We have now cut down the details presented in sections 3.1.1 on "Peak-to-peak amplitude and shape of seasonal cycle" and 3.1.2 on "Phase ellipses" and rewritten the section according to your suggestions.

*One of the conclusions drawn in the study is that the seasonal cycle of wetland emissions is the dominant driver of the seasonal cycle of d13C in the NH and tropics (see corresponding sentence in the abstract). However, this seems to be primarily based on the comparison between simulations E5_NS and E5_WETNS. Comparing simulations E432 and E432_EFMMNS would tell a different story, namely that a seasonal cycle in agricultural CH4 emissions could also perturb the d13C seasonal cycle. Which effect is closer to reality should be decided based on the comparison against observations and not just on the comparison between individual sensitivity runs. Note that in all simulations using EDGAR 5 (E5, E5_WETNS, E5_NS) the agriculture sector has no seasonal variability, too, and therefore it is clear that this sector cannot contribute to differences between these runs. The conclusion drawn on page 12, line 280 (and again in the abstract and conclusions section), is therefore not valid. The main difference between simulations E5_WETNS and E5_NS (at least in the southern hemisphere) is that in E5_WETNS biomass burning emissions have a seasonal cycle but those of E5_NS have not. Comparing these two simulations tells a lot about the influence of biomass burning emissions (see e.g. panel for latitude band 0-30°S in Fig. 2), but this is hardly addressed in the paper.*

We apologize for the badly written results in this section. We agree with the reviewer that EFMM and biomass burning emissions also contribute and have now corrected the analysis of the results including the effect on those emissions.

P12 L280 is now modified as: The similar behaviour of SIM_NS and SIM_E5_WETNS indicates that wetlands are the largest individual source driving the $\delta^{13}$C seasonal cycle, apart from sinks.

Abstract: We found that wetland emissions are an important driver in the $\delta^{13}$C seasonal cycle in the Northern Hemisphere and Tropics, such that the timing of the $\delta^{13}$C seasonal minimum is shifted by ~90 days at 60° N--90° N from the end of the year to the beginning of the year when seasonality of wetland emissions is removed. The results also showed that in the Southern Hemisphere Tropics, emissions from fires contribute to the enrichment of $\delta^{13}$C in July--October. We also found that EFMM emissions in EDGAR v4.3.2 cause a strong depletion of $\delta^{13}$C during spring due to its seasonality.

Conclusion: Based on the simulation results, wetland emissions are found to be the largest individual CH$_4$ source driving the $\delta^{13}$C seasonal cycle, apart from atmospheric sinks. Wetland emissions are the key, especially in determining the timing of $\delta^{13}$C minimum peaks in the NH. Comparisons using EDGAR v5.0, v4.3.2, and additional emission scenarios removing the seasonal cycle of emissions from enteric fermentation and manure management (EFMM) in v4.3.2 show that EFMM emissions in v4.3.2 are responsible for the unique $\delta^{13}$C depletion in the NH in spring.

*Section 3.3 describes seasonal cycles in the stratosphere. This section has several weaknesses. First, it is poorly connected to the rest of the paper focusing on the troposphere. How stratosphere-troposphere-exchange affects the seasonal cycles of CH4 and d13C would actually be an interesting topic, but the simulation setup does not allow addressing this question. Second, averaging the seasonal cycles over a large vertical range from 88 hPa to the top of the atmosphere does not seem a good idea, as there are probably significant differences in seasonality across this altitude range. The authors should rather focus on a single level (e.g. 80 hPa) or present vertical cross-sections/vertical profiles of the evolution in the stratosphere. Third, the discussion of the results is not convincing. For example, the authors argue that the strong amplitude of the seasonal cycles at the South Pole compared to the North Pole is due to the Brewer-Dobson circulation being stronger in the Southern Hemisphere, but actually the Brewer-Dobson circulation is stronger in the Northern Hemisphere, see e.g. Rosenlof (1995; https://agupubs.onlinelibrary.wiley.com/doi/10.1029/94JD03122).*

We apologize for the confusion of results from the stratosphere and the unfortunate mistake about Brewer-Dobson circulation. Initially we included the results from stratosphere to the paper, because we thought it may be interesting for the reader even though the paper is focusing on troposphere. We have now removed the section on stratosphere and instead we shortly address the topic elsewhere.

Lines 521-525: Other than the tropospheric sinks, the stratosphere-troposphere exchange also affects to some extent (Wang et al., 2002), and therefore the stratospheric sinks of Cl and O(1D) could contribute to the tropospheric seasonality. In the stratosphere, the effect of emissions is negligible, and the seasonality is largely driven by the atmospheric sinks. This was true in our simulations as well. The chemical sinks strongly enrich the $\delta^{13}$C in the stratosphere. Therefore, the stratospheric air that returns to the troposphere can affect the tropospheric seasonality of $\Delta\delta^{13}$C in mid and high latitudes.

Lines 551-552: The focus of the study was the troposphere. Nevertheless, the tropospheric cycles are affected by the stratosphere-troposphere exchange, and this calls for further studies.

*Section 4.1 needs to be rewritten and shortened as well. Rather than discussing the results in the context of other studies, it basically presents a review of existing literature. This is not what I expect from a discussion section. Section 4.2 is slightly better but could also be shortened.*

We apologize for the weak discussion of the results in sections 4.1 and 4.2. Those sections are now corrected to meet the expectations of a discussion section.

*The concept of phase ellipsoids should be better explained in the introduction section or somewhere else in the paper. Without this it is difficult to follow the discussion in section 3.1.2. Why do sinks produce a line but emissions an ellipsoid? Actually, to my knowledge only seasonally changing emissions produce an ellipsoid.*

We apologize for the unclear explanations regarding the phase ellipse. When there is not seasonality in surface fluxes, and only atmospheric sinks are considered, the phase ellipse would form a straight line, as reactions to $^{12}CH_4$ and $^{13}CH_4$ is expected happen at the same time. Thus, there is not time lag between $CH_4$ mixing ratio and inverse of $\delta^{113}C$ cycles. However, when the seasonality of surface emissions (and also soil sink) is included, the ratio of total CH4 fluxes to $^{13}CH4$ fluxes varies, as different source sectors have different seasonality and isotopic signature values, and therefore a phase ellipse forms an ellipse-like shape. We agree with the reviewer that the written expressions were not clear enough. We have now explained the concept of the phase ellipsoids better and rephrased the text in the section to be clearer.

Lines 296-306: The seasonal cycle of $\Delta\delta^{13}C$ with respect to the $\Delta CH_4$ cycle can be examined with a so-called phase ellipse (Bergamaschi et al., 2000; Allan et al., 2001). The phase ellipses, where $\Delta\delta^{13}C$ are plotted against $\Delta CH_4$, contain the same information as the time series (Fig. 2), but they provide better visualisation of the phase difference between the two. Fig. 3 shows phase ellipses from the simulations using different emission fields at 30° latitudinal bands. The length of the major axis of the ellipse represents the amplitude of the seasonal cycle, and eccentricity represents phase differences. In addition, we examine the timing $d$ (DOY) when the shifted correlations ($p_s$) between $\Delta CH_4$ at a time $t$ and $\Delta\delta^{13}C$ at time $t + d$ are at minimum and maximum. This quantifies the differences in the timing of minimum (maximum) in $\Delta CH_4$ and maximum (minimum) in $\Delta\delta$ 13 C (Supplementary Figure S10). If $\Delta\delta^{13}C$ cycle is a perfect inverse of $\Delta CH_4$ cycle, the ellipse becomes a straight line with a negative slope. Such case would be when the $CH_4$ fluxes have no seasonal cycle and only the atmospheric sinks derive the seasonality of the mixing ratios. In that case, the $\Delta CH_4$ maximum (minimum) occurs simultaneously as $\Delta\delta^{13}C$ minimum (maximum), and we expect $d$ min = 0 and $d$ max = 366/2 = 183.

*Sections 2.2.1 and 2.2.2 presenting the anthropogenic and natural emissions are quite long. A lot of the information could be put in a table (category, data source, total emissions per category, amplitude of seasonal cycle, etc.) so that the text could be shortened by referring to the numbers in the table rather than listing all these numbers in the text.*

We agree with the referee that a table could be used and have followed the recommendation. We have now moved the Supplementary Table 3 with some modifications to the main paper. Please see Table 1.

*What is the basis for the selection of the three observation sites ALT, NWR and SPO? Data availability? Distance from sources? Would there be other stations that could have been looked at? Why were they not considered?*

We thank the referee for this question. The selection of the stations was done by choosing stations that represent noticeable features. We have now provided results from all the other stations, where data are available, in the Supplementary. Please see Figures S4-S8. Note that those additional sites do not provide much further information.

*Figure 2: It would be good to highlight the line corresponding to SIM_E5 as a thick black line as a reference. The discussion in Section 3.1.1 should mainly refer to this line and discuss other simulations with respect to their deviations from this reference.*

We thank the referee for the suggestion and agree to do so.

*The article needs substantial editing for grammar:*

We apologize for the incorrect grammar and now we have done language editing for the article by a native English speaker.

In addition to these comments, we have taken care of all the small details listed at the end of your feedback.

*Page 1, Line 3: What do you mean by "examined"? This is a very vague term.*

This has been replaced by a word "compared".

*P1, L3: "Those includes" -> "Those include"*

This has been corrected.

*P1, L5: "in addition to other sources" -> in addition to other natural sources"*

This has been corrected.

*P1, L6: "global uniform value" -> "globally uniform value"*

This has been corrected.

*P1, L9: "in north of" -> "north of"*

This has been corrected.

*P1, L11: "timing of d13C seasonal minimum" -> "timing of the d13C seasonal minimum"*

This has been corrected.

*P1, L11, "days in 60°" -> "days at 60°"*

This has been corrected.

*P1, L14-15: "In light of this research, comparison to the observation" -> "These comparisons"*

This has been corrected.

*P1, L19: In what sense should the "proportion of biogenic to fossil based emissions" be revised? Please be more specific.*

We apologize for not being enough specific. This is now specified as: These comparisons to the observation showed that the seasonal cycle of EFMM emissions in EDGARv5.0 inventory is more realistic than in v4.3.2. In addition, the comparison at northern stations (north of 55° N) showed that modelled $\delta^{13}$C amplitudes are generally underestimated by 12–68 %, mainly because the model could not reproduce the strong depletion in autumn. This indicates that $CH_4$ emission magnitude and seasonal cycle of wetlands may need to be revised. Results from stations in northern latitudes (19–40° N) indicate that the proportion of biogenic to fossil-based emissions may need to be revised, such that a larger proportion of fossil-based emissions are needed during summer.

*P2, L32: "much studied" -> "studied intensively"*

This has been corrected.

*P2, L36: Why should these emission cycles depend on "political decisions"?*

We apologize for the badly formulated sentence. The new sentences are: Anthropogenic $CH_4$ emission seasonal cycles also have uncertainties. Although some countries report emission magnitudes to e.g., UNFCCC, often only annual values are reported, and emissions from e.g., rice paddies may not properly consider e.g., temperature dependencies and soil properties (Yan et al., 2009)

*P2, L52: "also warned that" -> "also cautioned that"*

This has been corrected.

*P3, L58: Sinks do not have a strong seasonality due to the KIE but simply due to the seasonality of OH (or Cl) radicals.*

We agree with the referee and have now corrected this sentence. We apologize for the badly formulated sentence.

*P3, L63: The d13C maximum was two months EARLIER not later than the CH4 minimum in those studies.*

We agree with the referee, and we apologize for the mistake. This has now been corrected.

*P3, L79: "that are a coarsening from" -> "corresponding to a subset of"*

This has been corrected.

*P3, L79: ", and vertical mixing was calculated" -> ". Convective vertical mixing was calculated"*

This has been corrected.

*P4, L86: reactions of CH4 with OH, Cl and O1D are chemical but not photochemical reactions.*

We agree with the referee, and we apologize for the mistake. This has now been corrected.

*P4, L90: It is not quite clear from the description whether the levels of Cl and O1D are prescribed in the stratosphere or whether the reaction rates are prescribed (which is not exactly the same). Please clarify.*

We apologize for unclear explanations. This has now been corrected by following sentence: The first order loss rates for the reactions with Cl and O(1D) are considered only in stratosphere, where the reaction rates are prescribed based on the atmospheric chemistry general circulation model ECHAM5/MESSy1 (Jöckel et al., 2006).

*P4, L117: I wouldn't say that the values in the stratosphere are "not important", but they are certainly "less important". Troposphere and stratosphere are linked through stratosphere-troposphere-ex- Change.*

We agree that the expression was not good. We have now reformulated the sentence: In this study, the focus is on the troposphere. However, we acknowledge that troposphere and stratosphere are linked through stratosphere-troposphere exchange (Wang et al., 2002), and we briefly return to the simulated concentrations in the discussion.

*P5, L132: "enriched 13CH4" -> "enriched in 13CH4". The seasonal cycle "is" dominated, not "are" domi- Nated.*

This has been corrected.

*P5, L135: What do you mean by "keeping the same annual totals"? The same as in 2010?*

We agree that the explanation was unclear. We mean that the totals for each year are kept as they are in the data but by applying the seasonality provided in the EDGAR. Hopefully, the new sentence in the article clarifies this: We calculated the seasonal cycle for each 1°×1° grid by applying the 2010 seasonality to other years, keeping the same annual totals as original for each year.

*P5, L143: "due to the differences" -> "due to the following differences"*

This has been corrected. See the sentence below in the next comment.

*P5, L143-146: These sentences are a bit confusing and could probably be formulated more elegantly.*

We have now formulated these better: The differences in the seasonality are mostly due to the following: In v4.3.2, the seasonality varies over latitude bands, while in v5.0, it varies by country for which information is available (Crippa et al., 2020). In addition, in v4.3.2 the same temporal profiles are used for all agricultural sectors, which is revised in v5.0 to better correspond each sector separately (Janssens-Maenhout et al., 2019; Crippa et al.,2020

*P5, L148: Delete "compared to v5.0".*

This sentence has been removed.

*P5, L150: "increasing trend" -> "increasing emissions"*

This has been corrected.

*P7, L183: For some sources, spatially varying isotopic signatures were used following Feinberg et al.*

*(2018). It would be good to know in more detail how Feinberg et al. collected this information. Cer-*

*tainly, it was not generated "based on global chemistry-climate simulations with SOCOL".*

We apologize for the badly formatted sentence. This is now corrected in the text: Spatially varying isotopic signatures are used for EFMM, coal, oil and gas, wetlands, biomass burning and geological emissions. For EFMM, oil and gas, coal and biomass burning, we use the signatures from Feinberg et al. (2018). EFMM isotopic signatures from Feinberg et al. (2018) are based on the local ratio of C3 and C4 vegetation (Still et al., 2003) and emitted isotopic signature of livestock fed with diet C3 or C4 (Sherwood, 2016). Oil and gas isotopic signatures taken from (Feinberg et al., 2018) are based on country-level natural gas and oil signatures (Sherwood et al., 2016). For coal, we use the M-COAL version presented by Feinberg et al. (2018) and references therein, which is based on the coal rank and depth.

*Table 2: Replace "vary globally" by "varying globally" in the table caption.*

This has been corrected.

*P8, L184: Geological and wetland emissions should not be described together in a single sentence, be-*

*cause these are completely independent sources.*

This has now been corrected. The new sentences are: For geological emissions, the globally varying isotopic signatures are taken from Etiope et al. (2019). Wetland isotopic signatures are taken from Ganesan et al. (2018) and the values are based on observations characterising wetland ecosystems.

*P8, L199: What are the values proposed by Monteil and how do they differ from those of Thompson?*

*Why do you not list the values of Monteil also in Table 1?*

We thank you for the question and are glad to inform you that those value of Monteil has been updated for comparison to Table 2, please see it. (Please note that Table 1 in the previous article is now Table 2).

*P9,L202: It should probably be "In contrast" rather than "In addition".*

This has been corrected.

*P9, L203: I don't understand the statement "decimal values instead of rounded integers" at all (I can only guess what this means).*

We apologize for the unclear explanation. This is now better formulated by: In contrast, we found that the simulated seasonal cycles in $^{13}CH_4$ are extremely sensitive to the applied spatial distribution and the absolute values of source signature up to decimal level, especially with region of high emission magnitude (Supplementary Fig. S3).

*P9, L212: Why "In contrast"? NWR is also far away from sources and measures background air like SPO. I don't see the contrast.*

This has been corrected.

*P9, L213: "Rockies" -> "Rocky Mountains"*

This has been corrected.

*P10, L227: "we examined the seasonal cycle" -> "we examined the impact of the seasonal cycle"*

This has been corrected.

*P10, L252: "in 90°S – 60°S approximately" -> "at 90°S-60°S is approximately"*

This sentence has been removed because of rewriting the section.

*P12, L269: "in spring" -> "from spring"*

This sentence has been removed because of rewriting the section.

*P12, L274: "two maxima peaks" -> "two maxima"*

This sentence has been removed because of rewriting the section.

*P12, L275: "In south of" -> "South of"*

This sentence has been removed because of rewriting the section.

*P12, L277: "closer follows" -> "more closely follows"*

This sentence has been removed because of rewriting the section.

*P12, L282: shouldn't it be "increased" rather than "decreased"?*

You are correct. However, this sentence has been removed because of rewriting the section.

*P12, L296: "only one month" -> "only during one month"*

This sentence has been removed because of rewriting the section.

*P13, L305: "the strong effect" -> "the strong contribution"*

This sentence has been removed because of rewriting the section.

*P13, L308: "and magnitude of autumn" -> "and the magnitude of the autumn"*

This sentence has been removed because of rewriting the section.

*P13, L318: "against that of" -> "against those of"*

This has been corrected.

*P15, L342: On page 4 the KIE of the reaction with OH had a value of 1.004. Here it has a value of -3.98*

*per mil. How do these numbers relate?*

The KIE of the reaction with OH is 1.004. We have now revised the number presented in the paper and the correct number is –3.79 ‰. These numbers are related so that the value of –3.79T ‰ is calculated using the value 1.004. We have now clarified the explanation in the text, lines 310-311: Using the KIE of OH, resulted in a slope of -0.0023 ‰ ppb$^{-1}$corresponding to -3.79310‰. The corresponding value is obtained by multiplying the slope with the mean of the smoothed fit $CH_4$.

*P17, L408: I can't see in Figure 4 that the modelled amplitude is larger "mainly due to a deeper mini-*

*mum". The maximum seems to be equally overestimated.*

You are correct and now this has been corrected.

*P18, L429: "not only the cause" -> "not the only cause"*

This has been corrected.

*P18, L437: "depletion better than in ALT" -> "depletion better"*

This has been corrected.

*P18, L445: "reach at maximum" -> "reach a maximum"*

This has been corrected.

*P18, L446: "After DOY=75 that" -> "After DOY=75"*

This has been corrected.

*P18, L449: "reach its maximum" -> "reach their maximum"*

This has been corrected.

*P19, L466-467: I don't understand why the first sentence leads to the conclusion presented in the sec-*

*ond sentence. It should rather be the other way round: "Because" chemistry time scale is shorter than*

*transport d13C stabilized more quickly at mid- to high latitudes and not "therefore".*

This section has been removed.

*P19, L476: Remove the "Delta" and add a "C" after d13.*

This section has been removed.

*P20, L491: This sentence needs clarification. A factor 1.5 to 7 change in the amplitude of the seasonal*

*cycle of d13C due to a simple change of the isotopic signature (of what??) by 1 per mil sounds like a*

*huge effect, which could make any model simulations useless.*

These sentences have been corrected as follows: The seasonality of $\delta^{13}$C depend on the isotopic signature of the emissions. Changes in $^{13}CH_4$ emissions for 0.1 % could result in $\sim$1 ‰ differences in $\delta^{13}$C. Considering that $\delta^{13}$C seasonal cycle amplitude is 0.07–0.26 ‰ (Fig. 2), changing the isotopic signature for a few per mill can reduce or increase the $\delta^{13}$C seasonal cycle amplitude by 1.5–7 times, and the timing of minima/maxima by 7–130 days at 30° latitudinal bands (Supplementary Fig. S11).

*P21, L504: "within in a country" -> "within a country"*

This has been corrected.

*P22, L567: Here a "(reference)" is missing.*

This has been corrected.

*P24: L614-615: The last two sentences in this section contradict the previous sentence concluding that*

*the chlorine sink is not important.*

This has been corrected.

*P24, L624: "are reverse of deltaCH4 cycles" -> "are opposite to those of deltaCH4"*

This has been corrected.li

*P25, L633: "moel" -> "model"*

This has been corrected.

*P25, L633: "extend" -> "extent"*

This has been corrected.

---

## Author Comment (AC2)

**Authors' response to anonymous referee #2**

In the following, *referee's comments are in italic*, authors' responses in normal font.

We thank the anonymous referee #2 for providing the valuable feedback. We apologize that the current form of the paper did not meet expectations.

Please note that some of the Figure and Table numbers have changed due to modification.

Article:

Table 1 *new

Table 1 -> Table 2

Table 2 -> Table 3

Table 4 *new

Figure 5 *removed

Supplementary:

Table 3 * removed

Table 3 * new

Table 4 *new

Figure 4-> Figure 9

Figure 4 *new

Figure 5 *new

Figure 6 *new

Figure 7 *new

Figure 8 *new

Figure 5 -> Figure 10

Figure 6 *removed

Figure 7 *removed

*The manuscript as it stands contains a lot of detail in the text which does not add to scientific understanding and is hard to follow in places, which could be made more concise. In addition, the main aims of the simulations and conclusions could be better defined. For example, what are the major uncertainties in the methane budget and how/where can examining its seasonal cycles help reduce these uncertainites? What is new here compared to previous studies analysing the seasonal cycles of CH4 and d13C-CH4? There are also a variety of typos/grammar which require correcting.*

We apologize for the excessive amount of unnecessary information in the article. We have now reduced the number of details and rewritten some part of the article to better meet your expectations. The major uncertainties are not the sources itself but the magnitude of each source. Analysing $\delta^{13}$C seasonal cycles of various sources, e.g., anthropogenic (enriched with $\delta^{13}$C) vs biogenic (depleted in $^{13}$C) can tell us whether some of the components are underestimated or overestimated by inventories. We have now formulated the goals of the study and the motivations of used methods better in the Introduction section to clarify, what new this study brings. Please see Lines 71-89. Please find more information what is new in this study from the beginning of Reply to anonymous referee #1. We also apologize for the typos/grammar errors, and now we have had the text checked by native English speakers.

*L3: change to 'These include emissions' etc.*

This has been corrected.

*L8: The text says that the phase ellipses do not form straight lines due to emissions. However, later in the text one of the conclusions is that even with constant emissions, the phase ellipses are still not straight lines due to the influence of sinks other than OH.*

We apologize for the unclear expression that was misleading. We have now rephrased the sentence: Due to surface fluxes, the anti-correlations between CH$_4$ and $\delta^{13}$C are not perfect and experience large variation (p=-0.35 to -0.91) in north of 30° S.

*L36: Do the authors mean 'Reported inventories for anthropogenic based thermogenic and biogenic CH4 emission seasonal cycles mainly depend on political decisions' rather than the emissions themselves?*

We apologize for the unclear expression for this. We have now rephrased the text as follows: Anthropogenic CH$_4$ emission seasonal cycles also have uncertainties. Although some countries report emission magnitudes to e.g., UNFCCC, often only annual values are reported, and emissions from e.g., rice paddies may not properly consider e.g., temperature dependencies and soil properties (Yan et al., 2009).

*L55: A literature range for the magnitude of the soil sink would be useful here.*

We thank you for the feedback. We have now added this: The estimated magnitude of soil sink also varies; from bottom-up estimates is 11–49 Tg $CH_4$ yr$^{-1}$ and from top-down estimates 27–45 Tg $CH_4$ yr$^{-1}$ $^{-1}$ (Saunois et al., 2020, and references there in).

*L58: Rephrase the way it is written, it sounds like the seasonality of the sinks is due to the KIE which is incorrect.*

This has been corrected and the new sentence is: Sinks enrich the atmosphere in the $^{13}CH_4$, due to their kinetic isotopic effect (KIE), and they have a strong seasonality, mainly due to the seasonality of OH radical in troposphere, and Cl and O(1D) in stratosphere.

*L94: Is Crowley et al. 1999 the correct reference here (the only mention of a KIE for CH4+OH I could see in this paper puts it at ~1.005, based on Cantrell et al. 1990)? Apologies if I've missed something.*

We apologize for the unfortunate mistake of references here. We have now corrected it as: The kinetic isotopic effects (KIE) k($^{12}CH_4$)/k($^{13}CH_4$ )= 1.004 and 1.013 are used for $^{13}CH_4$ Oh and O(1D), respectively (Saueressig et al., 2001), and 1.066 (Crowley et al., 1999) is used for Cl.

*L100-105: Is the magnitude of the soil sink used in TM5 known (Tg/yr)? There is quite a large range in the literature, and the magnitude assumed would be relevant for the CH4 seasonal cycles based on text later in the manuscript.*

Thank you for your feedback. We have now rephrased the sentence and added information about the magnitude of the soil sink used in this study: Yearly totals are varying 32.7–33.8 Tg $CH_4$ yr$^{-1}$.

*L117: The d13C-CH4 values in the stratosphere could be relevant for your analysis if seasonal variations in stratosphere-troposphere transport influenced d13C-CH4 seasonal cycles in the troposphere. Is this something that has been considered or might be significant at higher latitudes?*

We acknowledge the stratosphere-troposphere exchange. However, this study focuses on the modelled values in the troposphere and therefore we don't address the topic of stratosphere much. However, the STE effect can be significant, but would need another study. Further discussion can be found from lines 521-525: Other than the tropospheric sinks, the stratosphere-troposphere exchange also affects to some extent (Wang et al., 2002), and therefore the stratospheric sinks of Cl and O(1D) could contribute to the tropospheric seasonality. In the stratosphere, the effect of emissions is negligible, and the seasonality is largely driven by the atmospheric sinks. This was true in our simulations as well. The chemical sinks strongly enrich the $\delta^{13}C$ in the stratosphere. Therefore, the stratospheric air that returns to the troposphere can affect the tropospheric seasonality of $\Delta\delta^{13}C$ in mid and high latitudes.

*L154: 'EFWW' should be 'EFMM'*

This has been corrected.

*L188: I'm not sure I understand exactly what the missing data in grid cells is here. Do the authors mean, for example, there are grid boxes where you have wetland emissions from LPX-Bern, but no isotopic data is available for that grid box from Ganesan et al.? Could this be clarified?*

We apologize for the unclear expressions and have now clarified this in the article: We first examined the filled values (grids with no initial value assigned) by applying the values from Monteil et al. (2011) and Thompson et al. (2018) (Table 2).

*P9: Consider combining section 2.3 and 2.5 (or move location of Table 2)? Table 2 appears in Section 2.3, but is not mentioned in the text until Section 2.5.*

We apologize for the misallocation of Table 2 (now Table 3).

*L245: Section 3.1.1: There is a lot of detail here that I'm not sure contributes to the science understanding gained from this paper, and most of which can be deduced from looking at Figure 2. I think it would be easier to follow if the amount of detail regarding e.g. exact lag periods, percentage change in seasonal cycle magnitudes between certain simulations, could be reduced, so that the text highlights the most important differences between the simulations that can be seen in Figure 2, and how this adds to our understanding.*

We apologize that this section did not meet expectations and we have now rewritten the section.

*L321: "The effect of the soil sink is small at these latitudes, so probably the seasonal cycle of d13C is preliminarily driven by the atmospheric sinks at these latitudes". I think a bit more information would be useful here. The ellipse produced for SIM_NS differs in gradient at all latitudes from the theoretical KIE line when only OH is considered. Why do the authors think this is if not the soil sink?*

We have rewritten this section. The phase ellipse at the SH high latitudes do not follow the theoretical KIE line, and that can be due to the effect of other atmospheric sinks, i.e., stratospheric Cl and O(1D), and horizontal long-range and vertical transport. However, we should not have discarded the effect of soil sink completely, and the sentence/section is rewritten as follows (please see lines 323-333): At latitudes south of 30° S, the SIM_E5 phase ellipses' eccentricities are very high, with strong anti-correlation compared to those in the NH (Fig. 3). This indicates that $\Delta CH_4$ and $\Delta \delta^{13}C$ are close to the perfect inverse phases, i.e., the seasonal cycles are330preliminary driven by the atmospheric sinks and little affected by the seasonal cycle of emissions. However, the phase ellipse from SIM_NS does not exactly follow the KIE line, indicating the effect of sinks other than OH, i.e., stratospheric Cl, O(1D) and soil sinks, and horizontal long-range and vertical transport.

*Figure 3 caption: The solid black line is described as 'the KIE line of SIM_NS, and considering only the OH sink', this is confusing. Later in the text (L340), I think is the correct description for the solid black line: 'the theoretical KIE line when only the OH sink is considered'.*

You are correct and this has now been corrected. We apologize for the mistake in the description.

*L325: "When there are seasonal cycles only in the atmospheric sinks…", I think should be "When seasonal cycles solely arise from seasonal variations in the atmospheric sinks…".*

This sentence has been rephrased.

*L326: If there are no emissions and no sinks affecting the seasonal cycle, then there would be no seasonal cycle? Could the authors clarify this?*

We apologize for the unclear text. It is clear, that there is no seasonality in the emissions and sinks. We have now rephrased this sentence: Such a case would be when the $CH_4$ fluxes have no seasonal cycle and only the atmospheric sinks derive the seasonality of the mixing ratios. In that case, the $\Delta CH_4$ maximum (minimum) occurs simultaneously as $\Delta \delta^{13}C$ minimum (maximum), and we expect $d_{min} = 0$ and $d_{max} = 366/2 = 183$.

*Section 3.1.2: Again there is a lot of detail here, but it's not clear what the take away message is to the reader. As for Section 3.1.1, this section should be made more concise.*

We apologize that this section did not meet expectations and we have now rewritten the section.

*L429-432: Could seasonal changes in transport patterns (horizontal or vertical from the stratosphere) also influence the seasonal cycle at Alert? Given the high latitude of Alert, the CH4 sink there is likely to be relative small all year compared to the tropics.*

It is possible that the transport patterns also influence the seasonality at Alert or other locations at high northern latitudes. However, we would need to do new research to investigate this, and this study is not focused on this topic and therefore unable to provide answers to this question. We shortly address stratosphere (lines 521-525 and 551-552).

Lines 521-525: Other than the tropospheric sinks, the stratosphere-troposphere exchange also affects to some extent (Wang et al., 2002), and therefore the stratospheric sinks of Cl and O(1D) could contribute to the tropospheric seasonality. In the stratosphere, the effect of emissions is negligible, and the seasonality is largely driven by the atmospheric sinks. This was true in our simulations as well. The chemical sinks strongly enrich the $\delta^{13}$C in the stratosphere. Therefore, the stratospheric air that returns to the troposphere can affect the tropospheric seasonality of Dd3 C in mid and high latitudes.

Lines 551-552: The focus of the study was the troposphere. Nevertheless, the tropospheric cycles are affected by the stratosphere-troposphere exchange, and this calls for further studies.

However, there are also other reasons such as the used wetland isotopic signature, see discussion lines 397-403: Although we have used a recently published spatial distribution of source signatures where available, there are still large uncertainties in the modelled $\delta^{13}$C values due to, e.g., the vegetation types, especially for the tropical wetlands (Ganesan et al., 2018; Fisher et al., 2011). We found that $\delta^{13}$C values in high northern latitude sites, e.g., Alert, were overestimated by the model. Isotope signatures in autumn in northern wetlands can be as low as -79.4 ‰ (Hornibrook, 2009), and using more negative isotopic signatures could lead to a better match with the observations. Wetland CH$_4$ emissions and their seasonality are large globally (Supplementary Fig. S1 and Table 1) and therefore any error in the source signature is expected to have a large impact on the seasonality of $\delta^{13}$C.

*L476: typo – 'Delta13'*

This has been corrected

*L487: The Discussion section does not discuss the results of the paper, but is instead a mostly a summary of literature available regarding seasonal variations in emissions.*

We apologize for the format of the discussion, and we have now edited this section to meet expectations.

*Line 557: reference missing?*

This has been corrected.

*L611-613: I disagree that results from this study support the conclusions of Gromov et al. (2018) who concluded a negligible tropospheric Cl sink for CH4, as the simulations are able to capture much of the seasonal variation in CH4 and d13C-CH4 at the South Pole. I think a further simulation including a tropospheric Cl sink and an assessment of its impact on the seasonal cycle at SPO would be required to back this statement up. As the authors point out, the highest Cl concentrations are anticipated to be in the tropics, and may not have a strong influence at SPO. Also, if the d13C-Ch4 seasonal cycle at SPO is mainly controlled by the atmospheric sinks, could the choice of KIE for 13CH4+OH used in TM5 influence results here? As the authors point out, there are 2 differing values in the literature.*

To address this argument and topic further, we would need to make new runs to investigate this topic. This study cannot provide an answer. It is true that the selection of KIE value has an impact on the results, but we didn't test different KIE values for reaction with OH. We used the value of KIE = 1.004 based on Saueressig (2001) as recommended in Burkholder et al. 2019. Please see lines 605-615 for changes.

*L631-632: To conclude that the comparison between observed and modelled d13C-CH4 at SPO suggests the emission seasonality in the model is at the right level, I think that Figure 4 needs to show an influence of emission seasonality on the d13C-CH4 seasonal cycle, which is not clear from the current plot. Perhaps if SIM_NS could also be plotted in Figure 4, this would show more of an impact?*

Thank you for your feedback. We have now added the simulation SIM_NS to the plot to provide more information.

*L633: typo, 'moel' should be 'model'*

This has been corrected